



# In-cloud characteristics observed in US Northeast and Midwest non-orographic winter storms with implications for ice particle mass growth and residence time

Luke R. Allen[1,a], Sandra E. Yuter[1,2], Declan M. Crowe[2,b], Matthew A. Miller[2], and K. Lee Thornhill[3]

[1]Center for Geospatial Analytics, North Carolina State University, 27695, Raleigh, NC, USA
[2]Department of Marine, Earth, and Atmospheric Sciences, North Carolina State University, 27695, Raleigh, NC, USA
[3]Analytical Mechanics Associates, 23666, Hampton, VA, USA
[a]Current Affiliation: Department of Meteorology, Stockholm University, 10691, Stockholm, Sweden
[b]Current Affiliation: Center for Disaster Research and Education, Millersville University, 17551, Millersville, PA, USA

**Correspondence:** Luke R. Allen (luke.allen@misu.su.se) and Sandra E. Yuter (seyuter@ncsu.edu)

**Abstract.** The spatial distribution of surface snowfall accumulation is dependent on the 3D trajectories of ice particles and their residence times through regions of ice mass increases and decreases. We analyze 42 non-orographic, non-lake effect winter storms in the Northeast and Midwest United States from the Investigation of Microphysics and Precipitation for Atlantic Coast-Threatening Snowstorms (IMPACTS) and Profiling of Winter Storms (PLOWS) field campaigns. In situ aircraft measurements

(1 Hz, ∼100 m horizontal distance) yield key data on vertical air motions, $RH_{ice}$, and number concentration. When suitable airborne radar data are available, we sort the in situ measurements by distance from cloud radar echo top.

90% of updrafts (vertical air motion $\geq 0.5\,\mathrm{m\,s^{-1}}$) were $\geq 1.2\,\mathrm{km}$ across. Measurements obtained within 3 km of cloud echo top were twice as likely (14% versus 7%) to have vertical velocities capable of lofting precipitation-sized ice compared to points sampled at lower levels. Below the near cloud top generating cell layer, most of the storm volume has $RH_{ice} \leq 95\%$

consistent with sublimation.

Rather than precipitation-ice growth within broad areas of vertical air motions, observations indicate that ice growth in these storms primarily occurs episodically within layers of overturning cloud-top generating cells with scales $\leq$ a few km. Below the generating cell layer, conditions for ice growth are rarer, and the ice particles usually either persist or shrink during most of their descent. The observed distributions of ambient in-cloud conditions provide benchmarks for evaluations of winter storm

model output.

## 1 Introduction

The 3D trajectories of precipitation-size ice particles through winter storms control the amounts and spatial distribution of surface snowfall accumulation. Blowing snow at the surface can further modify accumulations at local scales which we will ignore for this study. Quantitative precipitation forecasts of snowfall accumulations often have large uncertainties of 100%

or more (Novak et al., 2008, 2014, 2023; Greybush et al., 2017; National Weather Service, 2018). Vertical air motions (w) and relative humidity (RH) control where and when hydrometeors are nucleated, and grow or shrink in size. In this study, we





examine aircraft in situ data from two recent winter storm field projects, the Investigation of Microphysics and Precipitation for Atlantic Coast-Threatening Snowstorms (IMPACTS, 2020-2023; McMurdie et al., 2022) and the Profiling of Winter Storms (PLOWS, 2009-2010; Rauber et al., 2014). These field campaigns sampled several dozen winter snowstorms in the Northeast

and Midwest United States. The geographic regions encompass parts of the Appalachian Highlands, Atlantic Coastal Plain, and Interior Plains (Fenneman, 1916). Our focus is on storms in which neither lake-effect nor orographic processes are important. We analyze vertical air motions and RH with respect to cloud particle number concentrations measured by wing-mounted probes and radar-observed structures to provide situational context of the storm structure to yield insights into where ice mass increases are more and less likely to occur within the storm.

A key characteristic of vertical velocity within storms is that it is *spatial scale-variant*. In order to be clearly interpreted, observed and modeled vertical velocity values need to be accompanied by their associated spatial scale. On synoptic ($> \sim 2000\,\mathrm{km}$) and larger scales, the atmosphere is very close to hydrostatic balance, meaning that vertical air parcel accelerations are negligible and the mean vertical velocity is reasonably close to $0.0\,\mathrm{m\,s^{-1}}$. In winter storms, there can be gradual ascent on the order of a few $\mathrm{cm\,s^{-1}}$ when averaging along a warm conveyor belt (100+ km long; Browning, 1971). On convective and smaller

scales ($< \sim 10\,\mathrm{km}$), the atmosphere is usually not in hydrostatic balance and deviations away from a mean of $0.0\,\mathrm{m\,s^{-1}}$ are reasonable and often observed (Markowski and Richardson, 2010; Holton and Hakim, 2013). At these smaller scales, buoyancy, turbulence, and vertical pressure gradients can yield vertical motions of several $\mathrm{m\,s^{-1}}$ or more. Figure 1 uses vertical velocity measurements from a single flight leg during IMPACTS to illustrate that as the horizontal scale of measurements is increased from 0.1 km to 10 km, the maxima and minima in vertical velocity get closer to $0\,\mathrm{m\,s^{-1}}$. At a given spatial scale,

vertical air motions within winter storms, with the exception of orographic and lake-effect snow storms, are usually weaker than whose within deep convection (e.g., Blyth et al., 2013).

Microphysical properties of hydrometeors are time-integrated. The "microphysical pathway" is the time sequence of mass changes a hydrometeor undergoes as a function of the sequence of relative humidity (RH) and temperature environments it moves through along its trajectory through the storm. When the timescale of snow falling to the surface is increased, the

lengthening of the snow particle trajectory yields more time for advection by horizontal winds and for particle growth and/or shrinkage processes to occur prior to the particle reaching the surface. The spatial distribution of surface precipitation is highly sensitive to the time between when a cloud particle reaches precipitation size and begins to fall and when it reaches the surface (e.g. Smith, 1979; Colle and Mass, 2000; Colle and Zeng, 2004; Colle et al., 2005; Lackmann and Thompson, 2019). This duration has been called the cloud "delay time" and the "residence time" by different authors (e.g., Feingold et al., 1996;

Barstad and Smith, 2005; Smith, 2006; Janiszeski et al., 2023). We will use *residence time* in this article.

The residence time is a function of the starting altitude where the particle first grows to precipitation-size and starts to fall and the effective fall speeds of the particle along its trajectory. Effective fall speed is the sum of the terminal fall speed of the particle in still air at the relevant air pressure and the vertical air motion. Updrafts will decrease effective fall speed and increase residence time while downdrafts will do the opposite.

To move precipitation-size ice upward, the vertical air motion has to be greater than the fall speed. Typical observed fall speeds of precipitation-size ice (diameter $\geq \sim 0.2$ mm) are $1\,\mathrm{m\,s^{-1}} \pm 0.5\,\mathrm{m\,s^{-1}}$ (Fitch et al., 2021). The median fall speeds of





unrimed aggregates, rimed particles and graupel vary dependent on surface wind speed and turbulence. The classic relationships between snow fall speed and particle sizes were measured in nearly still air and temperatures close to 0°C (Locatelli and Hobbs, 1974). Fall speed distributions are broadened by turbulence. In high turbulence and at low temperatures (< -13 °C),

the sensitivity of fall speed to ice precipitation particle size essentially disappears (Garrett and Yuter, 2014). Based on these observations, we infer that upward air motions of at least 0.5 m s$^{-1}$ are required to loft most precipitation-size ice particles. For a typical warm front sloped at a grade of 1/300 (Markowski and Richardson, 2010, p. 122), the horizontal velocity of air impinging on that front would need to be at least 150 m s$^{-1}$ in order for the mean vertical velocity caused by the upglide over the front surface to reach 0.5 m s$^{-1}$. Because such fast horizontal air velocities do not occur in the troposphere, some

combination of frontogenetical circulation, buoyant accelerations, turbulence, and wave motions would be required to induce upward motion reaching 0.5 m s$^{-1}$ in winter storms.

Most previous work on characterizing the vertical velocity characteristics within non-orographic winter storms used wind profilers and Doppler radars to estimate vertical air motions at horizontal scales from ~0.01 to ~1 km. Estimation of the vertical air motions from airborne and ground-based vertically-pointing sensors is complicated by the varying fall speeds of

precipitation within the radar resolution volume (Gossard, 1988, 1994; Gossard et al., 1990; Rosenow et al., 2014) and by airborne radar pointing angle errors (e.g., Rauber et al., 2017).

Cronce et al. (2007) used a ground-based 915-MHz wind profiler to sample vertical velocities in three winter storm cases in the central and southern United States. The profiler was positioned for each storm to measure bands of enhanced radar reflectivity on the north side of cyclones. The vertical resolution of the profiler was 105 m. The profiler had a half-power

beamwidth of 9°, so the horizontal resolution at 2, 4, and 6 km altitudes was 310, 620, and 940 m, respectively. Cronce et al. (2007) focused their analysis on periods when enhanced reflectivity bands passed overhead. Within 9 heavy reflectivity features (maxima in signal to noise ratio > 7 dB) they sampled 1515 total data points across roughly 360 minutes of data sampling. Their measured vertical velocities ranged from -4.3 to 6.7 m s$^{-1}$, and 35% of their measured vertical velocities exceeded 1 m s$^{-1}$ (Cronce et al., 2007, their Fig. 14). The overall mean vertical velocity in their measurements within precipitation bands was

0.6 m s$^{-1}$. Oue et al. (2024) used ground-based vertically-pointing Ka-band radar with a vertical gate spacing of 15 m to characterize updraft velocities within four snow storms over eastern Long Island, NY. The radar half-power beamwidth was 0.32°, corresponding to a horizontal resolution at 2, 4, and 6 km altitudes of 11, 22, and 33 m, respectively. They found that updrafts, defined as upward Doppler velocity (*vertical air motion + particle fall speed*) ≥ 0.4 m s$^{-1}$, were mostly < 20 s in the time-height data, corresponding roughly to < 500 m in horizontal scale (Oue et al., 2024, their Fig. 5).

Rosenow et al. (2014) used airborne W-band radar data obtained by the NCAR C-130 aircraft to characterize vertical air motions in three Midwest United States winter storms, with a focus on the comma head region on the north side of the cyclone and on cloud-top generating cells. Their radar data had a 15 m vertical range gate spacing and a 0.7° beamwidth corresponding to a horizontal resolution at 2, 4, and 6 km distances from the aircraft of 24, 48, and 72 m, respectively. Within generating cells in the highest ~1.5 km of cloud radar echo, they found maximum vertical motions between 1 and 2 m s$^{-1}$ (Rosenow et al.,

2014, their Figs. 8 and 10). Below the generating cells, they found much weaker vertical motions within updrafts, on the order of 0.1-0.2 m s$^{-1}$ (Rosenow et al., 2014, their Figs. 8 and 10). Rosenow et al. (2014) also sampled discrete cells of elevated



convection above a 1 km deep rain layer. The base of the elevated convection was ~4-5 km below echo top, on the south side of the comma head region of the cyclone. Within the elevated convection, updrafts with peak velocities as strong as $7\,\mathrm{m\,s^{-1}}$ on 1 km horizontal scale were found (Rosenow et al., 2014, their Fig. 19). Rauber et al. (2017) used the HIAPER Cloud Radar, an

airborne W-band radar, to sample a winter storm with heavy snow in the northeast United States. There were updrafts sampled in that case as strong as $5\,\mathrm{m\,s^{-1}}$ (when accounting for $\sim1\,\mathrm{m\,s^{-1}}$ particle fall speeds) at 1 km horizontal scale and 1 km vertical scale associated with Kelvin-Helmholtz waves and generating cells (Rauber et al., 2017, their Fig. 10). The Kelvin-Helmholtz waves and generating cells were usually no more than 1-2 km wide.

Winter storms that yield snowfall over coastal plain population centers have less commonly been the focus of field campaigns

as compared to orographic winter snow storms (e.g., Stoelinga et al., 2003; Houze et al., 2017; Tessendorf et al., 2019). Vertical air motion distributions have been documented for cumulus environments (e.g., LeMone and Zipser, 1980; Yang et al., 2016; Qin et al., 2023), but this has yet to be done for winter storms. We fill this gap with a comprehensive analysis based on data from the 42 research flights during the IMPACTS and PLOWS field campaigns which sampled surface snow-producing storms. These two field projects are the primary research aircraft in situ data sets for winter storms with snow reaching the surface in

the Northeast and Midwest US. The vertical velocity and RH distributions presented here provide new insights into typical in-cloud conditions in these types of storms as well as a set of benchmarks for evaluating model simulations of extratropical cyclones in the Northeast or Midwest United States.

## 2 Data and methods

### 2.1 In situ measurements of vertical velocity and relative humidity

During IMPACTS, the in situ platform for storm penetration measurements was the NASA Airborne Science Program's P-3 Orion based at the Wallops Flight Facility. Cloud properties and ambient conditions were measured on 40 total science flights in 2020, 2022, and 2023 (McMurdie et al., 2022). For this study, we use a subset of 30 IMPACTS flights that sampled storms with surface snowfall. Some of these storms included portions with freezing levels aloft and surface rain. For this study we examined flight legs that included surface snow. Vertical air motions measured in situ by aircraft probes are not affected by

nearby precipitation fall speeds. The 3D ambient wind field is the vector difference between the speed of the aircraft with respect to the earth and the speed of the air with respect to the aircraft (Lenschow, 1986). The spatial resolution of the in situ measurements is a function of sampling frequency and air speed.

Fast response, high precision in situ measurements of the ambient 3D winds along with atmospheric state parameters (static pressure and temperature) and aircraft data (position and altitude) were measured on the P-3 using NASA Langley's Turbulent

Air Motion Measurement System (TAMMS). For IMPACTS research data sampling (not transit legs), the typical airspeed of the NASA P-3 is $100\,\mathrm{m\,s^{-1}}$ and the in situ data are archived at 20 Hz, yielding a resolution of 5 m. The speed of the air with respect to the Earth is obtained with a 5-hole radome array arranged in a cruciform pattern to provide angles of attack and sideslip, combined with a Rosemount non-deiced temperature probe and pressure measurements corrected for position error. The inertial velocities, along with position and altitude data are provided by an Applanix 610. The raw data is collected



via a real-time data system at 100 Hz and then averaged down to both 20 Hz and 1 Hz final archive products which include
the 3D wind field. The TAMMS has been flown on the P-3 since before 2000. Extensive calibrations are done to account
for pressure defect, heading offset and the coefficients needed for the angles of sideslip and attack. The TAMMS data are
archived by the NASA Global Hydrometeorology Resource Center (GHRC) at https://cmr.earthdata.nasa.gov/search/concepts/
C1995869822-GHRC_DAAC.html (Thornhill, 2022).

Relative humidity was measured using an Edgetech Vigliant Model 137 chilled mirror hygrometer on the P-3, which has a
measurement uncertainty of ∼5% for RH with respect to ice (RH$_{ice}$). We will interpret measured RH$_{ice}$ < 95% as conditions
with ice shrinkage by sublimation, and measured RH$_{ice}$ > 105% as conditions with ice growth by vapor deposition. Points with
values in between are uncertain. In the 2023 IMPACTS deployment, two diode laser hygrometers (DLH; Diskin et al., 2002)
with different laser path lengths were used to precisely measure in situ humidity on the P-3, in addition to the standard chilled

mirror hygrometer. For the 2023 deployment in which there were multiple corroborating measurements of RH$_{ice}$, the chilled
mirror hygrometer and the DLH data were generally in close agreement. We will use corrected chilled mirror hygrometer data
in our RH$_{ice}$ analysis. We apply a bias correction for differences between the DLH and the chilled mirror hygrometer using a
linear regression. The chilled mirror hygrometer works by maintaining the temperature of a mirror at the dew or frost point so
that there is condensation on the mirror. The measured dew points (and thus relative humidities) can oscillate around the true

ambient value, especially in conditions with sharp gradients in RH, for example near cloud boundaries. We apply a bandstop
filter to the chilled mirror hygrometer data to remove oscillations on wave periods between 15 and 90 s. Additionally, the
chilled mirror hygrometer has a delayed response time compared to the DLH. Based on detailed comparisons between the
two sensors, we shift the chilled mirror hygrometer data earlier by 8 seconds. The chilled mirror hygrometer data are included
in the P-3 Meteorological and Navigation data, archived by NASA GHRC at https://cmr.earthdata.nasa.gov/search/concepts/

C1995868137-GHRC_DAAC.html (Yang Martin and Bennett, 2022).

For PLOWS, the National Science Foundation/National Center for Atmospheric Research (NSF/NCAR) C-130 was equipped
with both remote sensing and in situ instrumentation for 18 science flights in 2009-10 (Rauber et al., 2014). We used data from
the subset of 12 flights that sampled surface snow-producing winter storms. Similar to the TAMMS used in IMPACTS, the
C-130 was equipped with a gust probe located on the radome of the aircraft and an inertial navigation system which provided

measurements of the 3-D wind vector. The PLOWS data are archived at 1 Hz. The 1 Hz measurements correspond to about
100 m spatial scale. We use the version of PLOWS data in the official archive at https://data.eol.ucar.edu/dataset/113.063
(UCAR/NCAR - Earth Observing Laboratory, 2011).

The wind data from the two aircraft and campaigns can be tied together via an intercomparison made between the two back in
2001 as part of the NASA Transport and Chemical Evolution in the Pacific (TRACE-P) and NCAR's Aerosol Characterization

Experiment (ACE-ASIA). Two coordinated flights were made with multiple legs lasting between 10 and 60 minutes from the
marine boundary layer up into the free troposphere. Thornhill et al. (2003) compared the mean values obtained for the velocities
and temperature as well as the variances, spectra, fluxes and cospectra between the two wind measurement systems. The mean
values, variances, and power spectra of the vertical winds showed excellent agreement between the NASA P-3 and NCAR
C-130 systems with the primary difference being in the low-frequency portion of the spectra due to the autopilot. Although





there have been improvements since 2001 to the two data systems it is unlikely that the wind measurements obtained by the two aircraft would have fundamentally changed.

To analyze the data which have minimal influences from large changes in pitch and roll, we use only data sampled when the aircraft is flying straight and level flight legs, excluding data sampled during turns, rolls, ascents, and descents. We use the same definition of straight and level flight legs for both IMPACTS and PLOWS. The aircraft (P-3 or C-130) must have a

pitch angle between -2° and 2° and a roll angle between -4° and 4°. If there is a gap shorter than 5 s between any two straight and level flight legs, the two legs are joined together (the gap is also considered straight and level). We also exclude data from the transit flight legs between the aircraft base location and the targeted sampling region. We only include data sampled at air temperatures $< 0°\mathrm{C}$. Across all the science flights during IMPACTS that we analyzed, there were 39.6 hours (142,614 1-second samples) of in-cloud vertical velocity data along 430 straight and level flight legs. For PLOWS, there were 22.8 hours (82,107

1-second samples) of in-cloud vertical velocity data along 779 straight and level flight legs.

IMPACTS primarily sampled snow-producing storms in the Northeast United States, with a few flights sampling over the Midwest, while PLOWS primarily sampled over the Midwest United States (Fig. 2a-b). The vertical air motion distributions sampled during these projects are the best evidence of natural conditions within these types winter storms available.

Most of the IMPACTS and PLOWS sampling was done in the northwest quadrant of cyclones (Fig. 2c-d). Air mass bound-

aries (warm and/or occluded fronts) are often found in the northwest and northeast quadrants, which are generally associated with frontogenesis and strong vertical wind shear (which may be sufficient for Kelvin-Helmholtz instability). The warmer air mass above warm fronts also often contains local potential instability (Markowski and Richardson, 2010, p. 132). Release of instability near warm and occluded frontal surfaces likely results in a broader distribution of vertical velocity in the northwest and northeast quadrants, compared to the southwest and southeast quadrants. During IMPACTS, most of the sampling occurred

∼500 km or less from the low pressure center, whereas PLOWS more often sampled 500-1000 km away from the low pressure center (Fig. 2c-d).

## 2.2 Coordinated remote-sensing data from NASA ER-2

In order to get a more complete picture of the snowstorm and its surrounding environment, IMPACTS utilized two aircraft flying coordinated flight legs (McMurdie et al., 2022; Heymsfield et al., 2024). Joining the storm penetrating NASA P-3 was

the NASA ER-2 which flew above the storm and had several downward-pointing remote sensing instruments. Airborne radar data from the NASA ER-2 puts the NASA P-3 in situ observations into their vertical storm structure contexts. Backscattered echo can occur as long as sufficient-sized particles are present in a volume, irrespective of the RH. We use data from the NASA Goddard Space Flight Center's Cloud Radar System (CRS; McLinden et al., 2021), a W-band (94 GHz), polarimetric Doppler cloud radar on the ER-2. We use reflectivity and Doppler velocity data from the CRS during straight and level ER-2 flight

legs when the ER-2 and P-3 were well-coordinated, defined by $\leq 3$ km horizontal distance and $\leq 5$ min time gap between the two aircraft. The edge of CRS echo (where reflectivity $\approx$ -20 dBZ) provides an estimate of the cloud boundary. When the P-3 and ER-2 were well-coordinated and the P-3 was within CRS echo, the P-3 distance from echo top was calculated as the vertical distance from the P-3 altitude to the nearest above altitude where CRS reflectivity $<$ -20 dBZ within the column. The





### 2.3 Cloud probe data and cloud definition

In situ cloud sampling during IMPACTS was handled by the SPEC (Stratton Park Engineering Company) research group. For this study, we are utilizing data from a subset of their wing probes, including the 2-Dimensional Stereo probe (2D-S; Lawson et al., 2006) and Cloud Droplet Probe (CDP; Lance et al., 2010) to determine when the P-3 was flying in cloud. The 2D-S detected cloud and precipitation particles 100-2000 μm in diameter, and the CDP detected cloud particles 2-50 μm in diameter (McMurdie et al., 2022). When either probe measured a total number concentration ($N_T$) $\geq 10^{-3}$ cm$^{-3}$, we consider the P-3 to have been in cloud at that time. In the example shown in Fig. 5, changes in the 2D-S total number concentration correspond to when the P-3 was in regions outside of cloud and just above cloud echo (at ∼50 km along the flight leg), higher reflectivity and higher particle concentrations (at ∼75 km along the flight leg), and lower reflectivity and lower particle concentrations (at ∼115 km along the flight leg). NASA GHRC archives the 2D-S data at https://cmr.earthdata.nasa.gov/search/concepts/C1995868627-GHRC_DAAC.html (Bansemer et al., 2022) and the CDP data at https://cmr.earthdata.nasa.gov/search/concepts/C1997744632-GHRC_DAAC.html (Delene and Poellot, 2022).

From the NCAR C-130 during PLOWS, we use a combination of a CDP to detect cloud particles 2-50 μm in diameter and data from the 2-Dimensional Optical Array Cloud Probe (2D-C; UCAR/NCAR - Earth Observing Laboratory, 2024) to detect cloud particles 25-800 μm in diameter. Similar to IMPACTS, we consider the C-130 to have been in cloud when either the 2D-C or CDP measured $N_T \geq 10^{-3}$ cm$^{-3}$. For PLOWS, the 2D-C and CDP data are both archived at https://data.eol.ucar.edu/dataset/113.063 (UCAR/NCAR - Earth Observing Laboratory, 2011).

### 2.4 Updraft envelope definition

Prior studies of updrafts and downdrafts in convective clouds have defined and identified continuous envelopes of vertical velocities meeting given thresholds. For updrafts, LeMone and Zipser (1980) identified envelopes within the vertical velocity time series which continuously exceeded 0 m s$^{-1}$ for $\geq 500$ m and 0.5 m s$^{-1}$ for $\geq 1$ s (at least ∼100 m). Yang et al. (2016) used a 0.2 m s$^{-1}$ threshold to account for measurement uncertainty, and they removed the 500 m diameter requirement in order to capture smaller updrafts.

We use a similar method to Yang et al. (2016) to identify updraft envelopes, but we use a 0.5 m s$^{-1}$ threshold for updrafts to account for measurement uncertainty. This threshold also represents a lower-end estimate for the terminal fall speed of precipitation size ice particles (Garrett and Yuter, 2014; Fitch et al., 2021). While Yang et al. (2016) combined updraft envelopes that were separated by a distance $\leq 50$ m, we skip this step because our data are at 1-second intervals (Yang et al., 2016's data were at 0.04-second intervals), and the P-3's typical ground speed was roughly 100 m s$^{-1}$. Figures 5 and 7 show examples of updraft envelope identification for P-3 flight legs. The flight leg in Fig. 5 is through a layer with generating cells and the flight leg in Fig. 7 is ∼2 km below the cloud-top generating cells. This latter flight leg was entirely within cloud (2D-S total number concentration $\geq 10^{-3}$ cm$^{-3}$) and eight updraft envelopes were identified.





For each updraft envelope, we compute the integrated upward mass flux ($\phi_{upward}$ in $\mathrm{kg\,m^{-1}\,s^{-1}}$) using Eq. 1:

$$\phi_{upward} = \rho \bar{w} L_{updraft} \tag{1}$$

where $\rho$ ($\mathrm{kg\,m^{-3}}$) is the air density, $\bar{w}$ ($\mathrm{m\,s^{-1}}$) is the mean vertical velocity within the updraft envelope, and $L_{updraft}$ (m) is
the length of the updraft envelope. $\rho$ is calculated from the ideal gas law using the virtual temperature ($T_v$ in K) by Eq. 2:

$$\rho = \frac{p}{R_d T_v} \tag{2}$$

where $p$ (Pa) is the air pressure, $R_d$ is the dry air gas constant ($287\ \mathrm{J\,kg^{-1}\,K^{-1}}$). $T_v$ is calculated from the air temperature ($T$ in K) and water vapor mixing ratio ($q$, unitless) using Eq. 3:

$$T_v = (1 + \varepsilon q)T \tag{3}$$

where $\varepsilon$ is the ratio between the molecular mass of water and dry air ($\sim$0.61).

## 2.5    Potential sampling biases with in situ aircraft data

With aircraft-mounted, in situ instruments, we are only able to sample along lines through the storms, representing a tiny portion of the total 3D storm volume. PLOWS in particular and IMPACTS to an extent targeted regions where upward motions and ice mass growth were expected. Hence, their observed updraft distributions are *likely upper bounds on what would be*
*present over the entire storm volume*. For safety, regions with severe turbulence are avoided during research flight missions. Severe turbulence only occurred during one IMPACTS P-3 flight where strong Kelvin-Helmholtz waves were present over the Gulf of Maine on 29-30 Jan 2022.

     Measurements of updraft breadth by in situ instruments are subject to underestimation because the aircraft will usually not perfectly bisect an updraft. We can quantify this by considering an idealized case with a spherical updraft with diameter $D$ and
vertical velocity $\geq 0.5\ \mathrm{m\,s^{-1}}$. We assume that the aircraft's path through this spherical updraft is a straight line which misses the center of the updraft by a distance $h$. The updraft breadth measured by the aircraft is the length of this straight line, $L_{updraft}$, which can be calculated as:

$$L_{updraft} = \sqrt{D^2 - 4h^2} \tag{4}$$

In reality, the regions where vertical velocity $\geq 0.5\ \mathrm{m\,s^{-1}}$ in winter storms will be irregularly shaped. A sphere is a convenient
idealized shape for these calculations as it is rotationally symmetrical, and is likely a reasonable approximation for order 1 km updrafts within cloud-top generating cells. One could imagine an oblong updraft region associated with Kelvin-Helmholtz waves with the major axis close to vertical. We calculate the distribution of measured updraft breadths for many random aircraft passes through the idealized spherical updraft (Fig. 3). We assume that $h$ is distributed such that within the cross-section of the spherical updraft which contains the sphere's center and is perpendicular to the aircraft path, the number of random aircraft
passes within a region of the cross-section is proportional to the area of that region. In other words, the probability of the



aircraft passing within a distance $s$ of the sphere's center is:

$$P(h < s) = \frac{\pi s^2}{\pi(\frac{D}{2})^2} = \frac{4s^2}{D^2} \tag{5}$$

The cumulative distribution function (CDF) of $L_{updraft}$ is $P(L_{updraft} < \Lambda)$, where $\Lambda$ is the proportion of the actual updraft diameter in $(0, D)$ (Fig. 4). From Eq. 4, this can be rewritten as $P(\sqrt{D^2 - 4h^2} < \Lambda)$. Rearranging this gives:


$$P(L_{updraft} < \Lambda) = 1 - P(h < \frac{\sqrt{D^2 - \Lambda^2}}{2}) \tag{6}$$

After substituting Eq. 5 into Eq. 6 and simplifying:

$$P(L_{updraft} < \Lambda) = 1 - \frac{4(\frac{\sqrt{D^2-\Lambda^2}}{2})^2}{D^2} = (\frac{\Lambda}{D})^2 \tag{7}$$

This means that using aircraft in situ data, the probability of measuring an updraft breadth smaller than some proportion of $D$ is equal to the square of that proportion. For example, if an actual updraft is $1 \, \mathrm{km}$ across, then 25% of random aircraft passes

through that $1 \, \mathrm{km}$ updraft will measure the updraft to be $< 0.5 \, \mathrm{km}$ across. Similarly, since $0.7^2 = .49$, then 49% of random aircraft passes will measure that $1 \, \mathrm{km}$ updraft to be $< 0.7 \, \mathrm{km}$ across.

The updraft envelopes identified from in situ data following Sect. 2.4 will all represent *underestimates* of the real horizontal breadth of updrafts. Assuming spherical updraft shapes, then 25% of the measured updraft envelope lengths will be below half the actual updraft breadth (Fig. 4). If one multiplied the length of the measured updraft envelope by 3.2, then it would

overestimate the actual updraft for 90% of the updrafts. As a rough rule of thumb, multiplying the updraft envelope length by 3.2 will overcompensate for underestimates of updraft breath because the aircraft did not exactly bisect an updraft.

## 2.6 ERA5 reanalysis

We use ERA5 reanalysis data (Hersbach et al., 2020) to obtain the large-scale context for the in situ aircraft data. ERA5 data are output on a $0.25°$ grid globally at $1 \, \mathrm{h}$ intervals. The data are available on either a single level (e.g., 2-meter temperature,

10-meter wind components; Hersbach et al., 2023a) or on pressure levels (e.g., $800 \, \mathrm{hPa}$ temperature and wind components; Hersbach et al., 2023b).

To track surface lows, we used the algorithm from Crawford et al. (2021) with ERA5 mean sea level pressure (MSLP) data (Tomkins et al., 2024). The low-relative position of the P-3 was calculated from the surface low tracks for each IMPACTS case.

We used ERA5 data to characterize the omega and frontogenesis environments sampled by the P-3 during IMPACTS.

Frontogenesis describes the rate at which the gradient of a scalar field, e.g. potential temperature ($\theta$ in K), is changing with time in a parcel-following framework. We calculated 2D frontogenesis [$F_{2D}$ in $\mathrm{K} \, \mathrm{m}^{-1} \, \mathrm{s}^{-1}$, often displayed in $\mathrm{K} \, (100 \, \mathrm{km})^{-1} \, \mathrm{hr}^{-1}$] following Novak et al. (2004), who used a simplified form of the equations from Miller (1948):

$$F_{2D} = \frac{1}{|\boldsymbol{\nabla}\theta|} [-\frac{\partial\theta}{\partial x}(\frac{\partial u}{\partial x}\frac{\partial\theta}{\partial x} + \frac{\partial v}{\partial x}\frac{\partial\theta}{\partial y}) - \frac{\partial\theta}{\partial y}(\frac{\partial u}{\partial y}\frac{\partial\theta}{\partial x} + \frac{\partial v}{\partial y}\frac{\partial\theta}{\partial y})] \tag{8}$$

where $|\boldsymbol{\nabla}\theta|$ is the magnitude of the horizontal gradient of $\theta$ ($\mathrm{K} \, \mathrm{m}^{-1}$), and u and v are the zonal and meridional components of

the flow (in $\mathrm{m} \, \mathrm{s}^{-1}$), respectively. Positive frontogenesis is indicative of forcing for ascent associated with frontal circulations (Lackmann, 2011, p. 140).



## 3 Results

### 3.1 Vertical velocity measurements and context for individual flight legs

Two contrasting flight legs within and below generating cells for conditions with snow at the surface highlight key differences
in the vertical velocity structures and large scale instabilities. For the IMPACTS flight leg on 23 January 2023 1435-1445
UTC, cloud top varied between 5 to 7 km altitude (Fig. 5). As the aircraft flew in and out of cloud at 5 km altitude, cloud
particle concentrations varied from near zero outside of cloud to $10^{-1}$ cm$^{-3}$ in cloud. Updraft envelopes were frequent and
narrow in regions with generating cells with upward motion $\geq 0.5$ m s$^{-1}$ being broken up by intermittent measurements
$< 0.5$ m s$^{-1}$ (around 20-40 km, 70-80 km, and 125 km along leg distance). The horizontal scale of most updrafts with
magnitudes $\geq 0.5$ m s$^{-1}$ was only a few hundred meters across. Based on ERA5 reanalysis (Fig. 6), the flight leg was to
the northwest of and near a band of frontogenesis $> 1$ K $(100\,\mathrm{km})^{-1}\,\mathrm{h}^{-1}$ at 700 hPa. There was negative frontogenesis (i.e.,
frontolysis) present at the P-3 flight level with modest large-scale upward motion.

The IMPACTS flight leg on 23 January 2023 1535-1546 UTC was entirely within cloud (Fig. 7) and had cloud particle
concentrations near $10^{-2}$ cm$^{-3}$. Cloud top was close to 7.5 km for most of the leg. The P-3's flight altitude (3 km) was ~2 km
the below the layer with generating cells. At this altitude, the P-3 encountered few updraft envelopes stronger than $0.5$ m s$^{-1}$.
This flight leg was to the northwest of and near a band of weak frontogenesis at 700 hPa. Along flight level, weak frontogenesis
was present (Fig. 8a) and large scale upward motion is indicated just above flight level.

### 3.2 In-cloud environments sampled by IMPACTS and PLOWS

The distributions of ERA5 frontogenesis and omega for all the IMPACTS flight legs utilized in this study (Fig. 9) show that the
most commonly sampled environments had frontogenesis near 0 K $(100\,\mathrm{km})^{-1}\,\mathrm{h}^{-1}$ and omega near -0.4 Pa s$^{-1}$. Strong fron-
togenesis and strong large scale upward motions are outliers. The air temperature distributions of the in situ samples are shifted
to higher temperatures for IMPACTS as compared to PLOWS (Fig. 10) since climatologically winter surface temperatures in
the Midwest US tend to be lower than in the Northeast US. As a consequence, PLOWS near-cloud top storm environments in
the Midwest US tended to be colder than those for IMPACTS which were primarily in the Northeast US. PLOWS had more
than twice as many 100-m measurements as IMPACTS at temperatures $\leq$ -22°C, corresponding to polycrystalline and multiple
ice growth modes (Hueholt et al., 2022). Samples at air temperatures between -22°C and -8°C represent more than half the
samples from both IMPACTS (80,170 100-m samples) and PLOWS (45,927 100-m samples). In this temperature range, ice
growth mode is a function of both temperature and RH, with multiple growth mode at low supersaturations with respect to ice
($< 105\%$), tabular growth for RH$_{water} < 100\%$ and branched growth for RH$_{water} > 100\%$ (Hueholt et al., 2022).
Sorting the vertical velocity data by air temperature mainly illustrates the different aircraft sampling strategies and cloud top
temperatures between the IMPACTS and PLOWS projects (Fig. 11) rather than illustrating systematic structural differences
among storms. The vertical velocity distributions sorted by temperature categories corresponding to different ice growth modes
(Hueholt et al., 2022) are very similar to the overall project distribution for IMPACTS. For PLOWS, there is a higher incidence




of values $> 0.5\,\mathrm{m\,s^{-1}}$ at air temperature $\leq$ -22°C (11%) as compared to between -8 and -22°C (5%). The small number of
PLOWS samples at air temperature $>$ -8°C likely makes that distribution not representative.

## 3.3 Updraft envelope properties

We identified 2253 updraft envelopes $\geq 0.5\,\mathrm{m\,s^{-1}}$ during IMPACTS and 1079 updraft envelopes during PLOWS. The distribution of updraft envelope length is highly skewed (Fig. 12a). The vast majority of updraft envelopes were narrow, i.e. the updraft envelope threshold was only barely met for a brief time period (few hundred $\mathrm{m}$ distance). Median updraft envelope lengths and
intensities were 0.27 $\mathrm{km}$ and 0.75 $\mathrm{m\,s^{-1}}$ for IMPACTS and 0.24 $\mathrm{km}$ and 0.72 $\mathrm{m\,s^{-1}}$ for PLOWS. The mean length of updraft envelopes observed during both IMPACTS and PLOWS was 0.53 $\mathrm{km}$. Overall, 90% of updrafts envelopes were shorter than 1.2 $\mathrm{km}$. There does not appear to be a correlation between updraft envelope length and mean vertical velocity, that is wider updrafts are not necessarily stronger (Fig. 12a). In aggregate, the more numerous narrower updraft envelopes contributed more of the upward mass flux than the more sparse wider updraft envelopes (Fig. 12b). During IMPACTS, 65% of the upward mass
flux (Eq. 1) within envelopes meeting our criteria was contributed by updraft envelopes narrower than 2 $\mathrm{km}$. For PLOWS, this value was 64%.

## 3.4 Observed distributions of vertical velocity and $RH_{ice}$

### 3.4.1 General characteristics

In these winter storms, the in-cloud vertical air velocity is usually nearly zero. The distribution of in-cloud vertical velocity
measurements during IMPACTS was centered near $0\,\mathrm{m\,s^{-1}}$ (mean: $0.04\,\mathrm{m\,s^{-1}}$, median: $0.03\,\mathrm{m\,s^{-1}}$), while the distribution for PLOWS was centered at slightly higher values (mean: $0.12\,\mathrm{m\,s^{-1}}$, median: $0.09\,\mathrm{m\,s^{-1}}$). The in-cloud vertical velocity distributions show mean and median values of a few $\mathrm{cm\,s^{-1}}$ and that less than 10% of the measurements (9.1% for IMPACTS and 7.5% for PLOWS) are capable of lofting snow ($\geq 0.5\,\mathrm{m\,s^{-1}}$) on $\sim$100-meter horizontal scales (Fig. 13). It is likely that the slightly higher mean and median values in PLOWS as compared to IMPACTS relate to PLOWS aircraft specifically tar-
geting altitudes with generating cells whereas IMPACTS had an observation strategy that included sampling a variety of storm structures at multiple altitudes and temperatures. We found a broader distribution of in-cloud vertical velocities sampled during IMPACTS than during PLOWS (standard deviations of 0.56 and $0.38\,\mathrm{m\,s^{-1}}$, respectively; Fig. 13a). In-cloud downdrafts were less common than in-cloud updrafts, especially during PLOWS. Just 6.3% of in-cloud vertical velocity measurements during IMPACTS, and 2.6% during PLOWS, were $\leq$ -0.5 $\mathrm{m\,s^{-1}}$. Because the research flights during IMPACTS and PLOWS
targeted regions of likely snow growth, we expect the proportion of aircraft samples with vertical velocity $\geq 0.5\,\mathrm{m\,s^{-1}}$ to be an *overestimate* compared to the proportion of the entire storm volume with vertical velocity $\geq 0.5\,\mathrm{m\,s^{-1}}$.

### 3.4.2 Characteristics sorted by distance from cloud echo top

Since cloud top altitudes can vary by several $\mathrm{km}$ even within the same storm, and the layer with generating cells tends to follow the cloud top, we have found sorting by distance from cloud echo top to be more useful in interpreting physical processes than





sorting by air temperature (Fig. 14). Data presented in context of cloud echo top are only for IMPACTS P-3 flight legs which are well coordinated with the ER-2 (Sect. 2.2, Fig. 2), representing a smaller sample size than the entire in-cloud data points in Figure 13.

In-cloud points sampled within 3 km of CRS cloud echo top were about twice as likely (13.8% versus 6.5%) to have vertical velocity $\geq 0.5 \ \mathrm{m\,s}^{-1}$ than points sampled farther below cloud echo top (Fig. 14a). For downdrafts, 8.9% of points < 3 km below cloud echo top had vertical velocity $\leq -0.5 \ \mathrm{m\,s}^{-1}$, compared to 4.1% of points > 3 km below cloud echo top. These vertical velocity distributions are consistent with frequent occurrence of generating cells in the layer within 2 km of cloud echo top as reported by Rosenow et al. (2014), Plummer et al. (2014), and Rauber et al. (2015).

The distribution of $\mathrm{RH}_{ice}$ relative to distance from CRS cloud echo top (Fig. 14b) suggests that regions within 2 km of cloud top height are the primary regions of ice mass increases in these winter storms. Lower regions of cloud (> 2 km below cloud top) are more likely to be regions where ice mass is constant or decreasing. Given a measurement uncertainty in $\mathrm{RH}_{ice}$ of ~5%, ice growth likely occurs at measured $\mathrm{RH}_{ice}$ > 105%, and ice shrinkage likely occurs at measured $\mathrm{RH}_{ice}$ < 95%. It is uncertain whether points between $95\% \leq \mathrm{RH}_{ice} \leq 105\%$ are subsaturated or saturated. The median $\mathrm{RH}_{ice}$ value increases from ~95% at 2 km below cloud echo top height to nearly 105% in the closest 0.5 km below cloud echo top height (Fig. 14b). The distribution of $\mathrm{RH}_{ice}$ broadens closer to cloud echo top, and the mode of the $\mathrm{RH}_{ice}$ distribution shifts to higher values at altitudes closer to cloud echo top. Points with $\mathrm{RH}_{ice}$ > 105% were observed less than one-tenth of the time at all altitudes more than 2 km below cloud echo top height.

Closer to cloud echo top, higher number concentrations were more likely to be observed. The distribution of total number concentration measured by the 2D-S (particles 0.1 to 2 mm in diameter) also broadens closer to CRS cloud echo top (Fig. 14c) as compared to lower altitudes. The 90th percentile of total number concentration increases from $\sim 10^{-2} \ \mathrm{cm}^{-3}$ at 4 km below cloud echo top to over $10^{-1} \ \mathrm{cm}^{-3}$ within 1 km below cloud echo top. The median total number concentration is nearly constant at $\sim 6 \times 10^{-3} \ \mathrm{cm}^{-3}$ for heights more than 1 km below echo top. The distributions of $\mathrm{RH}_{ice}$ sorted by categories of cloud particle concentrations shows that higher $\mathrm{RH}_{ice}$ values are more likely in conditions with higher particle concentrations (Fig. 15).

The distributions of vertical air motions, $\mathrm{RH}_{ice}$, and particle number concentrations tell a physically consistent story. Layers with overturning generating cells near cloud echo top have more frequent vertical air motions capable of lofting ice, more frequent in-cloud conditions that are supersaturated with respect to ice, and more frequent conditions with higher particle counts. $\mathrm{RH}_{ice}$ is higher closer to cloud top because stronger upward motions are more prevalent than at lower altitudes. Number concentrations are higher because $\mathrm{RH}_{ice}$ is higher. Saturated $\mathrm{RH}_{ice}$ facilitates ice nuclei activation and preservation of small ice particles.





## 4 Discussion


The preferred locations of stronger upward motions near cloud top relates to the frequent occurrence of generating cells at these locations. Locations near cloud echo top are inconsistent with the locations of frontal surfaces which tend to be in the lower portions of the storm (Wallace and Hobbs, 2006).

Based on the IMPACTS observations, the widely held belief that typical non-orographic, non-lake-effect winter storms con-
tain broad areas of vertical air motions capable of lofting precipitation-size ice is a misconception. Lackmann and Thompson (2019) mistook observations of vertical air motions obtained in generating cell layers near cloud top at 1-2 km horizontal spacing (Rosenow et al., 2014; Rauber et al., 2017) as corroborating their model output of lofting of precipitation-size ice particles over broad ($\sim$25 km) horizontal scales upwind of banded snowfall (Lackmann and Thompson, 2019, their Figs. 4, 5, 12, and 13). Their two case studies focused on storms with strong frontogenesis [$> 5\,\mathrm{K}\,(100\,\mathrm{km})^{-1}\,\mathrm{h}^{-1}$ at 12-km grid spacing
at 700 hPa; G. Lackmann, personal communication] which are uncommon making these two examples unrepresentative of the larger snow-producing storm population. More realistically, Novak et al. (2008) presented a modeling study of a winter storm with a linear region of strong frontogenesis and enhanced radar reflectivity. Their simulation produced vertical velocities exceeding $\sim$0.6 m s$^{-1}$ at horizontal scales of only a few km wide near the region of strong frontogenesis (Novak et al., 2008, their Fig. 12c). A contributing factor in the confusion about the occurrence of strong near upright vertical air motions in winter
storms is the common practice of plotting vertical cross-sections with high vertical to horizontal aspect ratios that yield verti-
cally exaggerated plots which distort features such as sloping fronts making them appear much more upright than they actually are. Broad-scale upward motions strong enough to loft snow particles appear to be rare in non-orographic, non-lake-effect winter storms.

The aircraft observations from IMPACTS suggest that we need to revise our understanding of the relative roles of lift along
frontal surfaces versus overturning generating cells near cloud top in the formation of precipitation-size ice mass in mid-latitude winter storms. Layers of generating cells are usually neither resolved nor parameterized in current numerical forecast models with grid spacings of several km or more. This "error of omission" may have inadvertently overemphasized the roles of frontal surfaces and yielded model output that has vertical velocity and RH distributions that are biased higher than observed. There is a large body of work based on $\leq 1$ km scale observations on the processes within generating cells (e.g., Syrett et al., 1995;
Evans et al., 2005; Kumjian et al., 2014; Plummer et al., 2014; Rosenow et al., 2014; Cunningham and Yuter, 2014; Rauber et al., 2015; Plummer et al., 2015; Rosenow et al., 2014; Keeler et al., 2016b, a, 2017). It is high time that cloud top generating cells be adequately accounted for in operational forecast models.

## 5 Conclusions

We used airborne in situ measurements from the IMPACTS and PLOWS field campaigns in the Northeast and Midwest US
(Fig. 2) to characterize the distributions of in-cloud vertical velocity, the horizontal scales of updraft envelopes, and RH$_{ice}$ characteristics. Based on straight and level flight leg data corresponding to about 100-m spatial scale, our key results are:



- Most updrafts were narrow. 57% in-cloud updraft envelopes exceeding a threshold of $0.5\,\mathrm{m\,s^{-1}}$ were less than $300\,\mathrm{m}$ in breadth (median envelope lengths of $0.27\,\mathrm{km}$ for IMPACTS and $0.24\,\mathrm{km}$ for PLOWS). 90% of updraft envelopes were less than $1.2\,\mathrm{km}$ in breadth.

- In-cloud points within $3\,\mathrm{km}$ of cloud top were about twice as likely (14% versus 7%) to have vertical velocity capable of lofting precipitation-sized ice compared to points sampled further below cloud top.

- The more numerous narrower updrafts contributed more upward mass flux than the scarcer wider updrafts. About 65% of the upward mass flux was moved by updrafts less than $< 2\,\mathrm{km}$ across.

- Conditions for ice growth (measured $\mathrm{RH}_{ice} > 105\%$) were more likely to occur near cloud echo top, while much of
the in-cloud cloud volume more than $2\,\mathrm{km}$ below echo top had conditions for ice shrinkage (measured $\mathrm{RH}_{ice} < 95\%$). Higher number concentrations of particles 0.1 to $2\,\mathrm{mm}$ in diameter are more likely to be observed closer to cloud echo top than at lower altitudes.

These results show that the types of winter storms sampled by IMPACTS and PLOWS (extratropical cyclones, e.g., Nor'easters, Alberta clippers, and Great Plains cyclones) contain mostly weak vertical motions incapable of lofting precipitation-size ice.
Our findings are not applicable to orographic or lake-effect snow storms which superimpose additional forcings on extratropical cyclones. The finding based on in situ data that upward motions ($\geq 0.5\,\mathrm{m\,s^{-1}}$) are present in only small portions of the cloud volume and are most common in regions near cloud top generating cells extends and confirms the work of Rosenow et al. (2014) and Rauber et al. (2015) who used airborne radar data obtained during PLOWS. In addition to generating cells, Kelvin-Helmholtz waves can yield upward motions $\geq 0.5\,\mathrm{m\,s^{-1}}$ in some regions of winter storms. These manifest as sporadic,
small convective scale updrafts ($< 2\,\mathrm{km}$ across; e.g., Rauber et al., 2017).

Many current conceptual models assume $\mathrm{RH}_{ice} > 100\%$ in radar echo and that precipitation-size ice grows gradually and relatively continuously as it descends from generating cells near cloud top to the surface (e.g., Plummer et al., 2014, 2015; Rosenow et al., 2014, 2018; Finlon et al., 2022). In contrast, IMPACTS data sets on vertical air motions suggest that ice growth is likely to be more episodic than continuous. Most updrafts stronger than $0.5\,\mathrm{m\,s^{-1}}$ were $< 300\,\mathrm{m}$ across, with regions of near
zero or downward vertical motion in between. Ice particles likely grow within these narrow updrafts for a brief period of time (on the order of minutes) relative to their overall residence time in cloud (on the order of $1\,\mathrm{h}$ to fall $4\,\mathrm{km}$).

Even when one overcompensates for the sampling bias of updraft envelopes size by multiplying aircraft measured length by 3.2 (Sect. 2.5), the vast majority of updrafts potentially capable of lofting snow are $< 1\,\mathrm{km}$. The altitude where precipitation-size ice *first* forms and begins to fall is likely the primary factor in residence time in winter storms rather than subsequent
lofting. There is a lack of observational evidence of commonly occurring broad scale precipitation-size ice lofting in these storms ($25+\,\mathrm{km}$ wide updraft regions with vertical air motions $> 0.5\,\mathrm{m\,s^{-1}}$) as proposed by Lackmann and Thompson (2019).

Surface snowfall rates and accumulations depend on where within the storm precipitation-size particles form and conditions along their trajectories to the ground. Generating cells near the top of winter storms are key regions where cloud ice can grow to precipitation size by vapor deposition and riming (Kumjian et al., 2014). The precipitation particles falling from cloud top





are then advected horizontally by flows within the storm. Convergent flow may result in locally higher precipitation particle concentrations (Janiszeski et al., 2023), or sheared flow may "smear" ice streamers together (Tomkins, 2024). Maintaining conditions of supersaturation with respect to water requires stronger updrafts than does maintaining supersaturation with respect to ice. Below the layer of generating cells, vertical motions are weaker, the air is less likely to be supersaturated with respect to ice, and ice mass shrinkage via sublimation is more likely. Sublimation can be a self-limiting process, as it increases
the ambient RH which later particles fall through. But, vertical air motions and ambient RH do not completely describe the conditions immediately adjacent (few microns) to the surface of individual ice particles. When there is nonzero airflow around an ice crystal (such as when it is falling and/or advected by horizontal winds), ventilation can enhance vapor density at an ice particle's corners, increasing RH immediately adjacent to the particle above ambient values (Hallett and Mason, 1958; Keller and Hallett, 1982; Takahashi et al., 1991; Fukuta and Takahashi, 1999). Quantitative ventilation effects for the complex shapes of natural snow are poorly understood and hence often not accounted for in numerical models but may well turn out to be an important process in ice mass budgets (Wang, 2002; Bailey and Hallett, 2002).

Convective-scale overturning motions are important to produce upward air motion $\geq 0.5\,\mathrm{m\,s^{-1}}$. The small spatial scales of updrafts (often $< 300\,\mathrm{m}$) implies that numerical model grid spacing $\geq 1\,\mathrm{km}$ may be inadequate for realistically simulating cloud processes in winter storms (as found by Bryan et al., 2003, for deep convection). Gradual large-scale layer lifting within
winter storms can contribute to the destabilization of the environment and to the subsequent release of upright instability allowing for cloud-top generating cells to form (e.g., Xu, 1992; Schultz and Schumacher, 1999; Morcrette and Browning, 2006).

There are many potential avenues for future work with the data collected during IMPACTS. In order to further explore the ambient conditions associated with ice particle growth and shrinkage, the in situ humidity data need to be analyzed in the
context of observed ice crystal shapes (Hueholt et al., 2022). The Particle Habit Imaging and Polar Scattering (PHIPS) probe (Abdelmonem et al., 2011) provided high-resolution particle images during each IMPACTS deployment. The PHIPS images are of adequate resolution to discern ice particle shapes and degrees of riming, which will inform on the sequences of growth modes of the ice particles (Hueholt et al., 2022). Corresponding humidity data will indicate whether ice particles are actively growing where they were observed ($\mathrm{RH}_{ice} > 100\%$) or not. How much of the precipitation ice particle's residence time is
spent in subsaturated conditions with respect to ice is highly relevant to surface snow fall rates. Not adequately accounting for subsaturated conditions within winter storm volumes may be an important contributing factor in the persistently large errors in surface snow rate forecasts.

*Data availability.* All of the NASA IMPACTS data are archived by GHRC at https://ghrc.nsstc.nasa.gov/uso/ds_details/collections/impactsC.html (McMurdie et al., 2019). The NSF PLOWS 1-second flight-level data are archived by the UCAR Earth Observing Laboratory at
https://data.eol.ucar.edu/dataset/113.063 (UCAR/NCAR - Earth Observing Laboratory, 2011). ERA5 hourly data on pressure levels are available from the Copernicus Climate Data Store at https://cds.climate.copernicus.eu/datasets/reanalysis-era5-pressure-levels?tab=overview



(Hersbach et al., 2023b). ERA5 hourly single-level data are available from the Copernicus Climate Data Store at https://cds.climate.copernicus.eu/datasets/reanalysis-era5-single-levels?tab=overview (Hersbach et al., 2023a).

The specific data shown in each figure of this article are available at https://zenodo.org/records/14224688 (Allen et al., 2024).

*Author contributions.* LRA and SEY conceptualized the project. KLT processed and archived the TAMMS data from IMPACTS. LRA, SEY, DMC, and MAM contributed to the analysis methodology. LRA, DMC, and MAM wrote the analysis software. LRA created the visualizations with input from SEY and DMC. LRA prepared the manuscript and SEY edited the manuscript. DMC, MAM, and KLT contributed to the final stages of reviewing and editing.

*Competing interests.* The contact author has declared that none of the authors have any competing interests.

*Acknowledgements.* The development of the methodology, interpretation of the results, and visualizations benefited from discussions and correspondence with DelWayne Bohnenstiehl, Brian Colle, and Matthew Parker. Figures 5cd and 7cd were created using code written by Laura Tomkins. John Barrick assisted with design, integration, and calibration of the TAMMS used during IMPACTS.

This work was supported by the National Science Foundation (AGS-1905736), the National Aeronautics and Space Administration (80NSSC19K0354), the Office of Naval Research (N000142112116 and N000142412216), and the Center for Geospatial Analytics at North
Carolina State University. The IMPACTS project was funded by the NASA Earth Venture Suborbital-3 (EVS-3) program managed by the Earth System Science Pathfinder Program Office.



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



**Figure 1.** In situ vertical velocity measurements from IMPACTS for a single flight leg from 22:20 to 22:47 UTC on 19 Jan 2023 averaged to different scales. **(a)** 1-second averages ($\sim$100-meter horizontal scale for $100\,\mathrm{m\,s^{-1}}$ aircraft horizontal velocity). **(b)** 10-second averages ($\sim$1-km horizontal scale). **(c)** 100-second averages ($\sim$10-km horizontal scale). In each panel, the dashed horizontal line is at $0.5\,\mathrm{m\,s^{-1}}$ vertical velocity, and purple shading indicates where measured vertical velocity $\geq 0.5\,\mathrm{m\,s^{-1}}$. As the averaging period increases, maxima and minima in vertical velocity generally approach $0\,\mathrm{m\,s^{-1}}$, and the portion of the flight leg where vertical velocity $\geq 0.5\,\mathrm{m\,s^{-1}}$ decreases. This flight leg was at 3.5 km ASL, where the air temperature was roughly -10°C. There are coordinated ER-2 radar data for this flight leg which indicate that the P-3 flew through a layer of generating cells, with CRS cloud echo top heights between 4 and 6 km ASL.



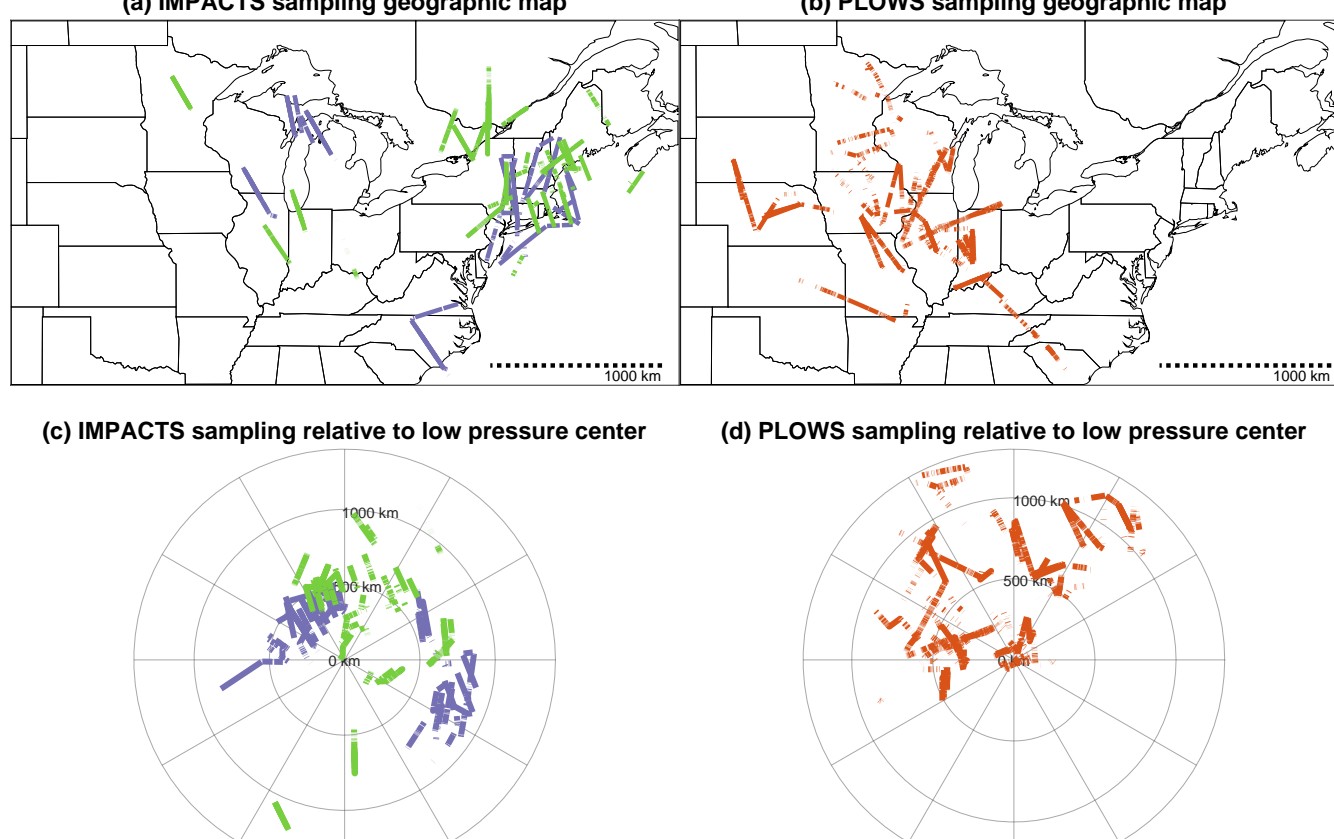

**Figure 2.** Geographic maps of in-cloud portions of level flight legs with air temperature $< 0°\mathrm{C}$ sampled during **(a)** IMPACTS by NASA P-3 (legs in close coordination with NASA ER-2 in green, P-3 only legs in blue) and **(b)** PLOWS by NCAR C-130 (red). Flight legs plotted relative to trackable low pressure centers for **(c)** IMPACTS and **(d)** PLOWS. Leg line segments are not continuous when the airplane sampled in and out of cloud.







**Figure 3.** Cross-section through the center of an idealized spherical updraft with diameter D in the xz-plane where the aircraft is traveling in the y-direction (shown by aircraft silhouette). Each dot represents one of 1000 randomly generated aircraft passes through the spherical updraft, such that the likelihood of the aircraft passing through a given subregion of the cross-section is proportional to the area of that subregion. Points are colored by the length of the aircraft's path through the sphere ($L_{updraft}$), relative to the actual diameter of the sphere ($D$).

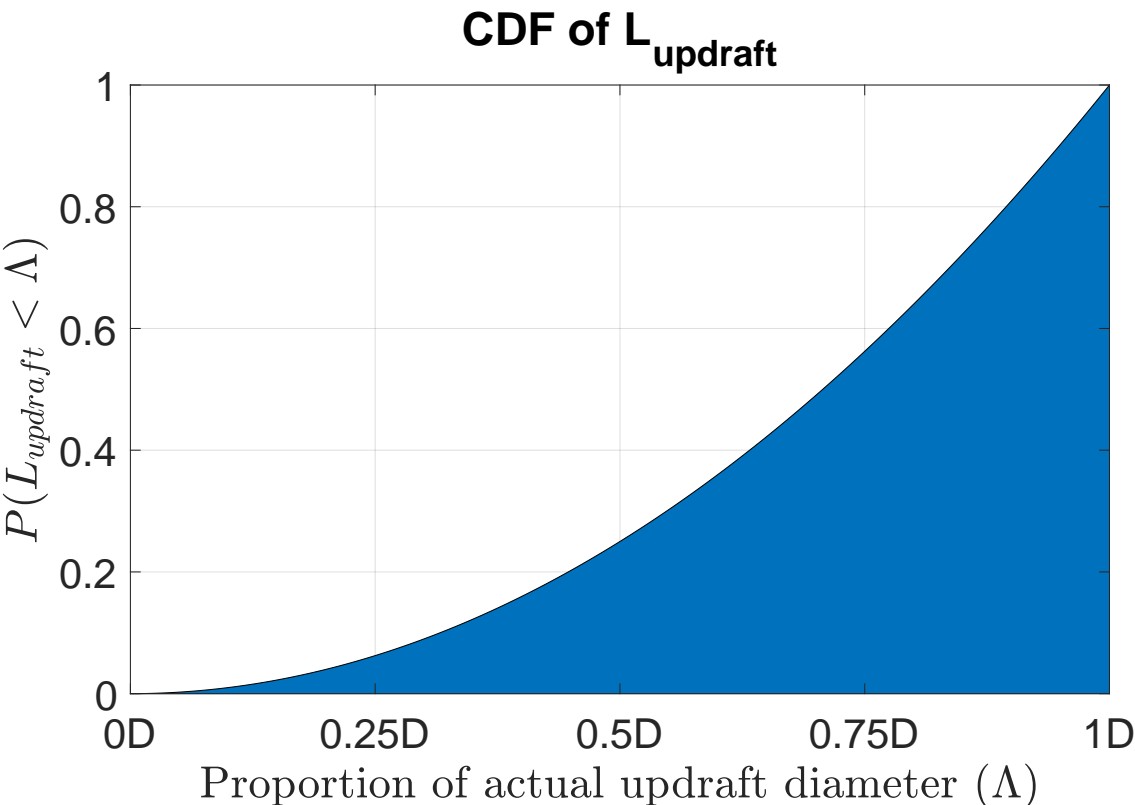

**Figure 4.** Theoretical cumulative distribution function (CDF) of the measured length of an updraft ($L_{updraft}$) by an aircraft randomly passing through a spherical updraft with diameter $D$, relative to $D$, according to Eq. 7. For each proportion $\Lambda$ of the actual updraft diameter $D$, the proportion of random aircraft passes measuring an updraft smaller than $\Lambda$ is shown (e.g., 25% of random aircraft passes measure an updraft smaller than $0.5D$).



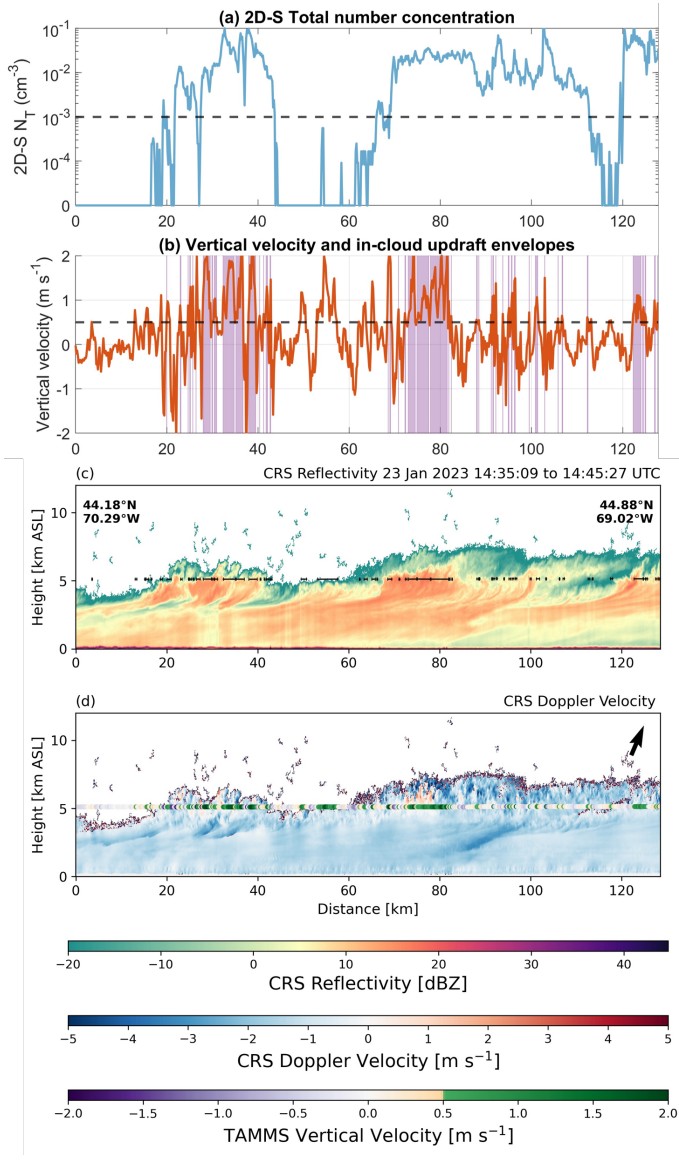

**Figure 5.** P-3 in situ and ER-2 radar data from for a well-coordinated flight leg segment during the IMPACTS mission on 23 January 2023 from 1435 to 1445 UTC. In this flight leg, the P-3 flew through a layer of cloud-top generating cells, and encountered multiple narrow updrafts $\geq 0.5 \, \mathrm{m \, s^{-1}}$. **(a)** Time series of 2D-S total number concentration (logarithmic scale). The in-cloud threshold ($10^{-3} \, \mathrm{cm^{-3}}$) is shown by a dashed horizontal line. **(b)** Time series of TAMMS vertical velocity. Dashed horizontal line at $0.5 \, \mathrm{m \, s^{-1}}$ indicates the updraft threshold definition. In-cloud updraft envelopes are indicated by purple shading. **(c)** Vertical cross-section of CRS reflectivity. P-3 flight leg segments where the TAMMS vertical velocity $\geq 0.5 \, \mathrm{m \, s^{-1}}$ are indicated by black horizontal lines bounded by vertical bars. **(d)** Vertical cross-section of CRS Doppler velocity, where positive values indicate upward motion. TAMMS vertical velocity is shown using colored points. Aspect ratio is 3:1 for panels **(c)**-**(d)**.

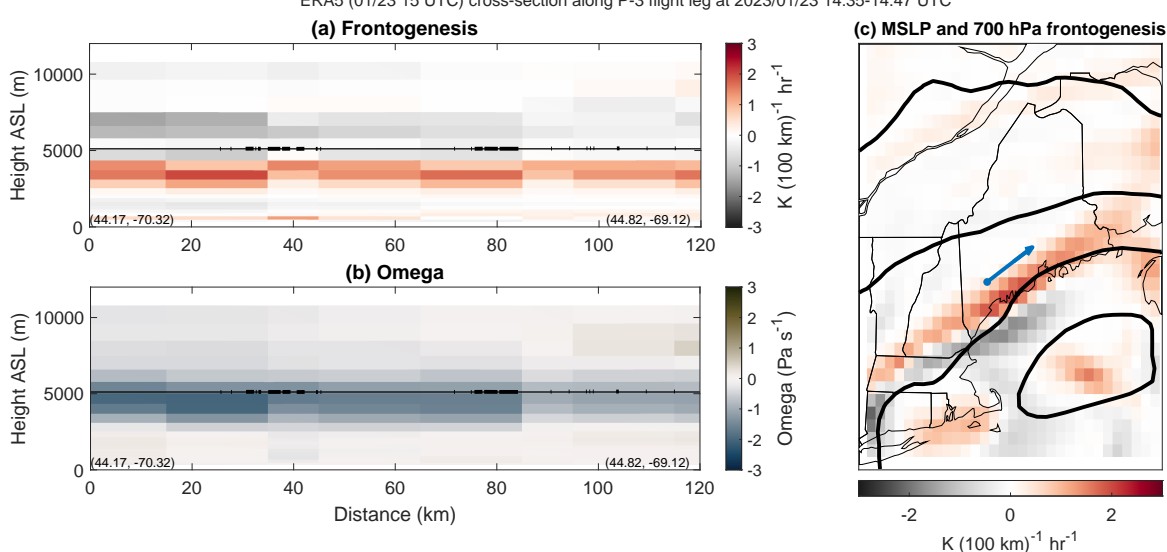

**Figure 6.** Large scale environmental setting for flight leg through generating cells corresponding to Fig. 5. The P-3 flew through multiple narrow updrafts $\geq 0.5 \, \mathrm{m \, s^{-1}}$ at an altitude with frontolysis and near the top of layer with stronger omega which was adjacent to a band with strong frontogenesis at 700 hPa. Vertical cross-sections (aspect ratio 3:1) of **(a)** 2D Frontogenesis [K $(100 \, \mathrm{km})^{-1} \, \mathrm{hr}^{-1}$] and **(b)** omega ($\mathrm{Pa \, s^{-1}}$, where upward motion is negative) obtained or calculated from ERA5 data along a cross-section corresponding to the flight leg shown in Fig. 5. In **(a)** and **(b)**, the P-3 flight level is indicated by a thin horizontal line, and observed in-cloud updrafts $\geq 0.5 \, \mathrm{m \, s^{-1}}$ are indicated by thick horizontal lines. **(c)** Map of the flight leg (blue), MSLP (contoured in black every 5 hPa), and 700 hPa frontogenesis [shaded, K $(100 \, \mathrm{km})^{-1} \, \mathrm{hr}^{-1}$].





**Figure 7.** As in Fig. 5, but for a well-coordinated flight leg on 23 Jan 2023 from 1535 to 1546 UTC when the P-3 flew through the middle of the cloud layer, about 2 km below the cloud-top generating cells. In this flight leg, the P-3 encountered far fewer updrafts $\geq 0.5\ \mathrm{m\,s^{-1}}$ than it did in the flight leg shown in Fig. 5.

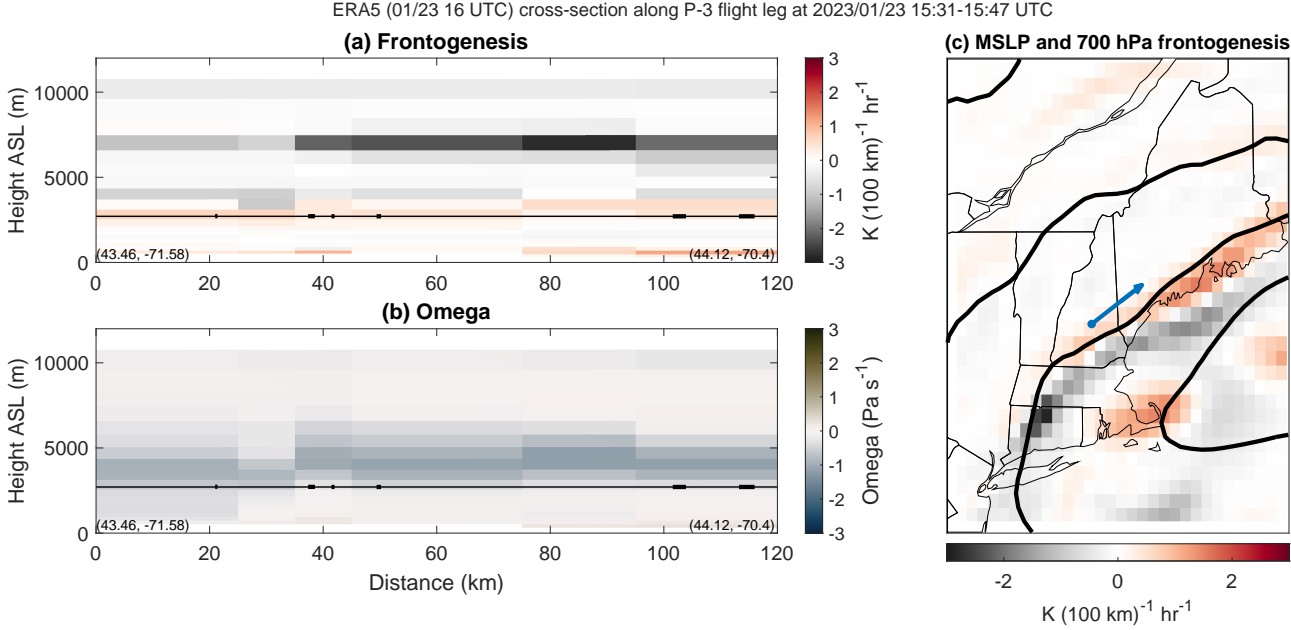

**Figure 8.** As in Fig. 6, but for flight leg several km below height of generating cells corresponding to Fig. 7. The P-3 flew through sparse updrafts $\geq 0.5\,\mathrm{m\,s^{-1}}$ at an altitude with near zero omega and weak frontogenesis adjacent to a band with weak frontogenesis at 700 hPa.



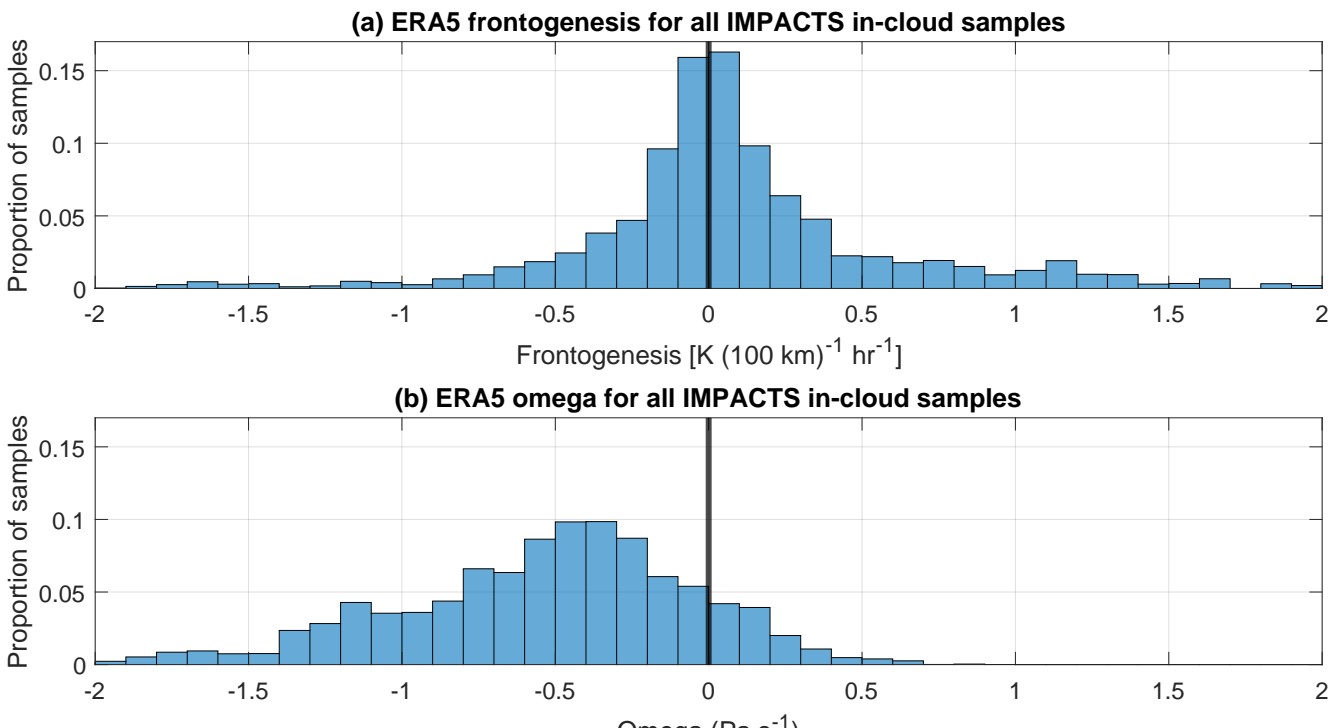

**Figure 9.** Distributions of ERA5 **(a)** frontogenesis $[\mathrm{K}\,(100\,\mathrm{km})^{-1}\,\mathrm{hr}^{-1}]$ and **(b)** omega $(\mathrm{Pa\,s}^{-1}$, where negative values indicate upward motion) for all IMPACTS P-3 in-cloud samples. For each IMPACTS sample point, the ERA5 frontogenesis and omega value at the nearest grid point at the nearest hour is taken. The 10th, 50th, and 90th percentiles of ERA5 frontogenesis for IMPACTS in-cloud sampled are -0.41, 0.03, and 0.79 $\mathrm{K}\,(100\,\mathrm{km})^{-1}\,\mathrm{h}^{-1}$, respectively. For omega, the 10th, 50th, and 90th percentiles are -1.20, -0.47, and 0.05 $\mathrm{Pa\,s}^{-1}$, respectively.





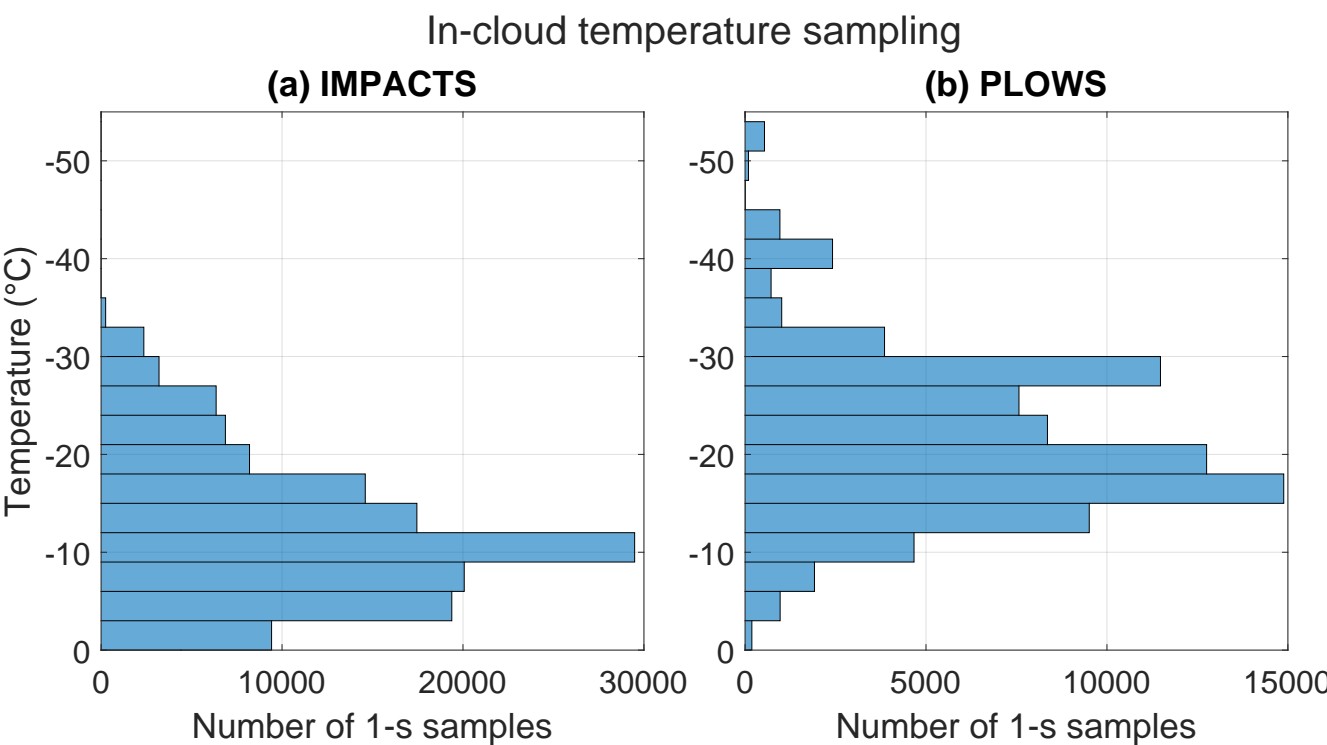

**Figure 10.** Distributions of in-cloud temperatures sampled during **(a)** IMPACTS and **(b)** PLOWS. Colder temperatures are at the top of each plot. The in-cloud environments sampled during PLOWS tended to be colder than those sampled during IMPACTS.



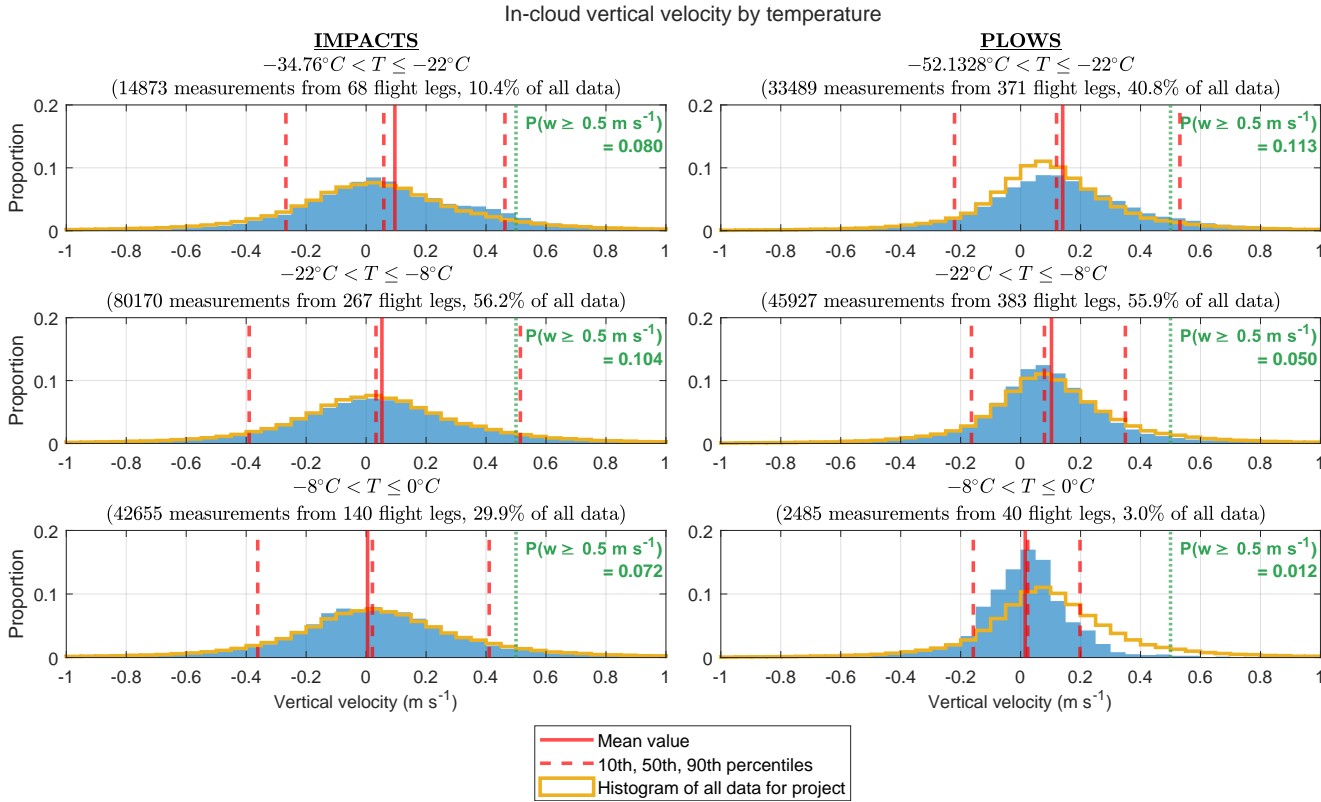

**Figure 11.** Histograms of in-cloud vertical velocity measurements taken on level flight legs at different temperature ranges, from colder temperatures at the top to warmer temperatures at the bottom. Distributions from IMPACTS are shown in the left column, and distributions from PLOWS are shown in the right column. Temperature ranges and sample sizes are indicated for each panel. Data are averaged over 1 s corresponding to about 100 m based on typical P-3 air speeds. Temperature ranges were chosen to correspond with different ice growth modes. Depending on the RH, polycrystalline or multiple growth can occur at temperatures < 22°C. Multiple, tabular, branched, or side branched growth can occur between -22 and -8°C. Multiple or columnar growth can occur between -8 and -4°C (Hueholt et al., 2022). Solid red lines show mean vertical velocity for each subset, and dashed red lines show 10th, 50th, and 90th percentile vertical velocity for each subset. In each column, gold lines show the distribution for all data from that project obtained in straight and level legs.





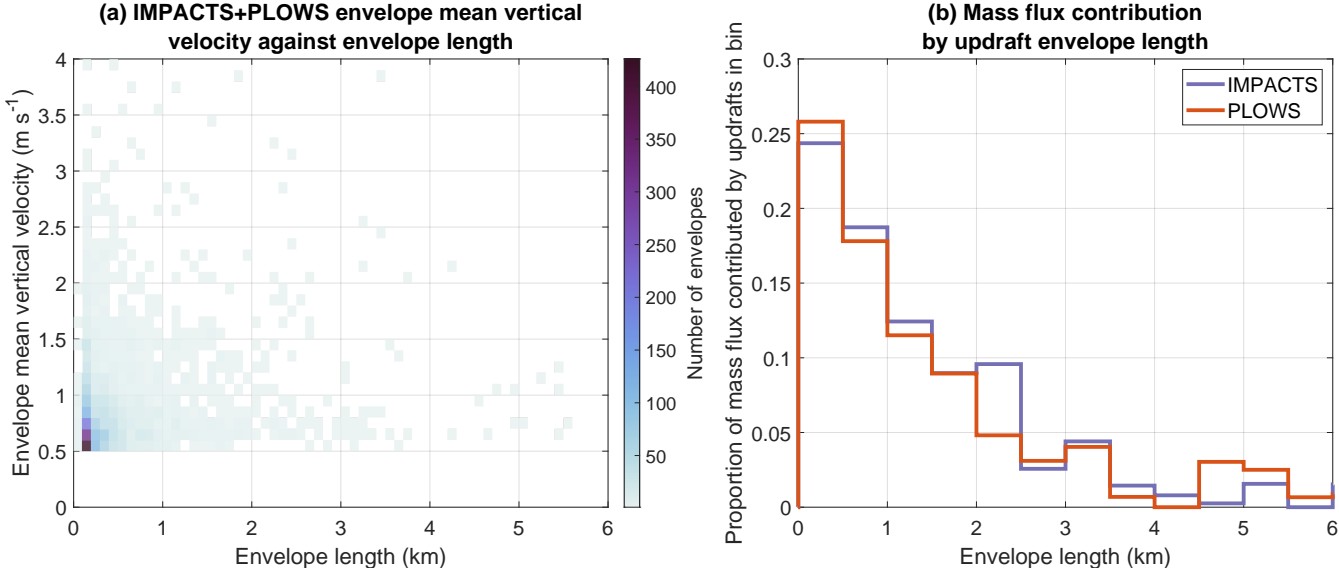

**Figure 12. (a)** 2-D histogram of updraft envelope mean vertical velocity against updraft envelope length, for IMPACTS and PLOWS combined. **(b)** The proportion of total upward mass flux contributed by updraft envelopes within 0.5-km length bins for IMPACTS and PLOWS separately.



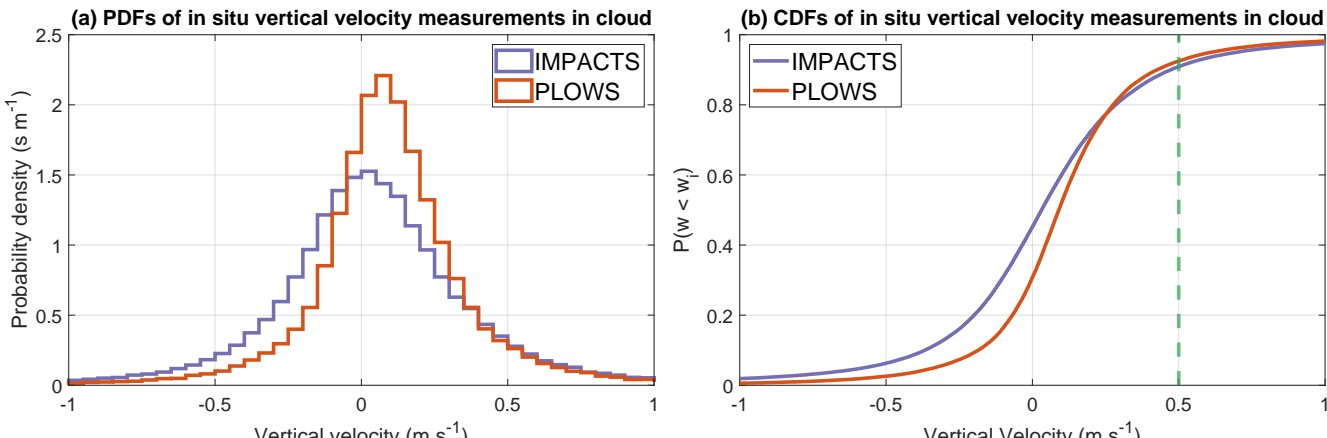

**Figure 13.** In-cloud vertical velocity measurements taken on level flight legs during IMPACTS and PLOWS on ~100-meter horizontal scale, **(a)** probability density function (PDF) and **(b)** cumulative density function (CDF). For both IMPACTS and PLOWS, > 90% of vertical velocity measurements on ~100-meter horizontal scale were < 0.5 m s$^{-1}$ [vertical dashed line in **(b)**].

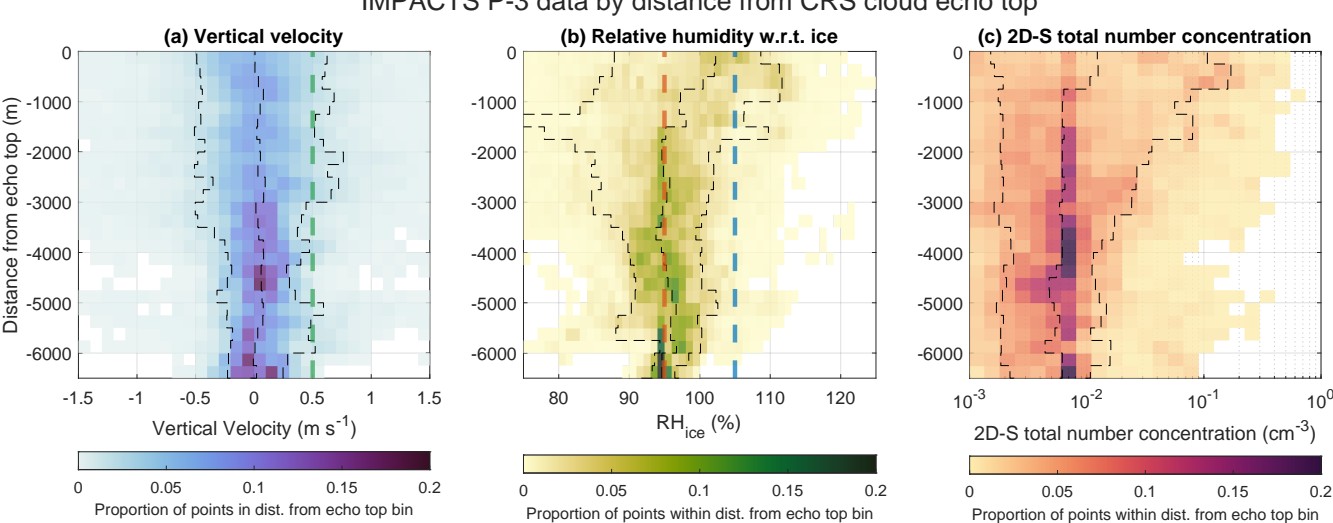

**Figure 14.** 2D histograms of **(a)** in situ vertical velocity, **(b)** $RH_{ice}$, and **(c)** 2D-S particle counts versus the P-3 distance below CRS cloud echo top height for in-cloud samples with air temperature $< 0°C$ during IMPACTS only. In all panels, the number of points in each bin is normalized by the total number of samples taken at a given distance below echo top. Black dashed lines indicate the 10th, 50th, and 90th percentile of vertical velocity as a function of distance below echo top height. For vertical velocity, the measurement uncertainty is $\sim 0.5\,\mathrm{m\,s^{-1}}$, and the vertical green dashed line is at $0.5\,\mathrm{m\,s^{-1}}$ vertical velocity. For $RH_{ice}$ the measurement uncertainty is $\sim 5\%$, the vertical brown line is at $95\%$ $RH_{ice}$, and the vertical blue line is at $105\%$ $RH_{ice}$.





**Figure 15.** IMPACTS NASA P-3 in situ $RH_{ice}$ and 2D-S particle counts from all IMPACTS in-cloud flight legs (~37 hours of data). Data samples are divided roughly into thirds. Histograms of $RH_{ice}$ values for **(a)** 2DS total counts $> 10^{-3}$ and $\leq 0.004\,\mathrm{cm^{-3}}$. **(b)** 2D-S total counts $> 0.004\,\mathrm{cm^{-3}}$ and $\leq 0.008\,\mathrm{cm^{-3}}$. **(c)** 2D-S total counts $> 0.008\,\mathrm{cm^{-3}}$. At most temperatures, $RH_{ice}$ from 100% to 105% corresponds to multiple growth mode where there is *co-occurrence* of growth of different ice shapes by vapor deposition (Bailey and Hallett, 2004, 2009). 2D-S probe counts all (liquid and ice) particles between 100-2000 $\mu$m size.