# Peer review of "In-cloud characteristics observed in US Northeast and Midwest non-orographic winter storms with implications for ice particle mass growth and residence time"

_EGUsphere, 2024_

## Referee Comment (RC1)

Review of "In-cloud characteristics observed in US Northeast and Midwest non-orographic winter storms with implications for ice particle mass growth and residence time", by Allen and coauthors, egusphere-2024-3808.

Characterizing the horizontal scales and vertical motions in wintertime precipitating clouds, snow generation in snowbands and generating cells are an important research topic. This study aims to address these areas using a comprehensive data set from three IMPACTS field campaigns (east coast and Midwest US) and the more limited data from the PLOWS (Midwest US) campaign. The main takeaways for me are that the horizontal scales of updrafts are small compared to the broader regions of snowfall, vertical motions are more likely near cloud top, the updrafts are primarily associated with generating cells and snowbands, and the $RH_i$ is close to 100%. What are not considered are the weaker updrafts, <0.25 m/s, that might well support much of the ice particle growth. A model could be used to estimate how much of the total water vapor mass upward flux is due to sub 0.5 m/s updrafts.

**General Comments**

Please comment on whether ice particle growth can largely be ignored outside of the strong updraft regions. Whether there is greater contributions to the total storm precipitation from the strong updrafts versus the weaker but more expansive regions.

Given the vertical velocity measurement accuracy, you are not measuring <20 cm/s updrafts. These can be significant in providing an upward flux of moisture for particle growth downwards.

Lines 220-221. precipitation size ice particles. According to the AMS Glossary of Meteorology, frozen precipitation: *Any form of precipitation that reaches the ground in frozen form, that is, snow, snow pellets, snow grains, ice crystals, ice pellets, and hail.* With 0.5 m/s threshold, how much precipitation would you be missing (percentage wise). Lines 423-424 These results show that the types of winter storms sampled by IMPACTS and PLOWS (extratropical cyclones, e.g., Nor'easters, Alberta clippers, and Great Plains cyclones) contain mostly weak vertical motions incapable of lofting precipitation-size ice.). Lines 423-424 These results show that the types of winter storms sampled by IMPACTS and PLOWS (extratropical cyclones, e.g., Nor'easters, Alberta clippers, and Great Plains cyclones) contain mostly weak vertical motions incapable of lofting precipitation-size ice. Please comment on the size range of particles that may be lofted at 0.5 m/s. This would provide an upper limit for the sizes that may be lofted.

I suggest adding a figure showing the distribution of temperatures sampled for IMPACTS and PLOWS

Should the measured updrafts be called flight segments (which they are) or envelopes (how do you know)?

Are the TAMMS inlets subject to icing and therefore error?

Lines 450-456. This is superfluous and should be removed.

**Minor Comments**

37 gravity waves

44. timescale of snow. From when. From its nucleation through to ...?

47. begins to fall. It's always falling during its growth, because it has a terminal velocity. It may not fall through the updraft. So, fall>fallout

51. "fall" to "fallout"

Fall speed should be "terminal velocity". fall speed is the net (updraft velocity-terminal velocity).

197. Should be University of North Dakota and NCAR

202. Fig. 5. Note in the text that this figure will be discussed in more detail later

Fig. 5 If you're using all channels of the 2D-S probe, because of uncertainty in the sample volume of the particles in the bins for the small particles, it's an overestimate because some of the particles may be drops, shattering affecting the concentration and uncertainties in the sample volume for the small particles.

314. $RH_{Water}$ >100% is extremely unlikely if not impossible, especially at these vertical velocities.

379-380. Saturation with respect to water not ice

399-407 This is an excellent paragraph

444. and aggregation

Figure 11 multiple ice growth. A better description is needed. Figure 11: <-22C

450-456. This is  superfluous and should be removed.

463-470. This part of the paragraph is superfluous and should be removed.

---

## Author Comment (AC1)

Reviewer comments
Author responses
Text from revised manuscript

**General notes from authors to all reviewers**:
In the original submission, we used air temperatures from the P-3 Meteorological and Navigation data (Yang-Martin and Bennett, 2022) in Figs. 10 and 11, and in the $RH_{ice}$ calculation for Figs. 14 and 15 (now 16). Upon further consultation with the instrument scientists, it was recommended to use the TAMMS temperature data because of their better calibration quality (Thornhill, 2022). Between the submission of the paper and work on revising the manuscript there was also an update to the NASA data archive containing a new version of the TAMMS vertical velocity values for IMPACTS flights in 2020. These new values slightly changed the vertical velocity distributions shown in Figs. 11, 13, and 14, as well as the updraft envelope properties shown in Fig. 12. As a result of these data revisions, we have updated the specific values, for example for distribution percentiles to our results and the aforementioned figures. Our main conclusions are unchanged. A comparison of the original submission and revised figures are below. Each of the changes in the text are also noted below.

We have added Figure 15, the joint frequency distributions of vertical air motion and RHice to address a reviewer comment and to clarify the relationships shown in Figure 14.

[Figure]

**Figure 15.** Joint frequency distributions of 100-m horizontal scale $RH_{ice}$ and vertical velocity (a) < 3 km below CRS echo top, and (b) ≥ 3 km below CRS echo top for the well-coordinated flight-legs during IMPACTS. The color scale in each panel is normalized by the total number of measurements in the panel. The sample size (number of measurements) is indicated in the upper-left corner of each panel.

Submitted version of Figure 10

[Figure]

**Figure 10.** Distributions of in-cloud temperatures sampled during (**a**) IMPACTS and (**b**) PLOWS. Colder temperatures are at the top of each plot. The in-cloud environments sampled during PLOWS tended to be colder than those sampled during IMPACTS.

Updated version of Figure 10

[Figure]

**Figure 10.** Distributions of in-cloud temperatures sampled during (**a**) IMPACTS and (**b**) PLOWS. Colder temperatures are at the top of each plot. The in-cloud environments sampled during PLOWS tended to be colder than those sampled during IMPACTS.

[Figure]

**Figure 11.** Histograms of in-cloud vertical velocity measurements taken on level flight legs at different temperature ranges, from colder

[Figure]

**Figure 11.** Histograms of in-cloud vertical velocity measurements taken on level flight legs at different temperature ranges, from colder

Submitted version of Figure 12

[Figure]

**Figure 12. (a)** 2-D histogram of updraft envelope mean vertical velocity against updraft envelope length, for IMPACTS and PLOWS combined. **(b)** The proportion of total upward mass flux contributed by updraft envelopes within 0.5-km length bins for IMPACTS and PLOWS separately.

Updated version of Figure 12

[Figure]

**Figure 12. (a)** 2-D histogram of updraft envelope mean vertical velocity against updraft envelope length, for IMPACTS and PLOWS combined. **(b)** The proportion of total upward mass flux contributed by updraft envelopes within 0.5-km length bins for IMPACTS and PLOWS separately.

Submitted version of Figure 13

[Figure]

**Figure 13.** In-cloud vertical velocity measurements taken on level flight legs during IMPACTS and PLOWS on ~100-meter horizontal scale, (a) probability density function (PDF) and (b) cumulative density function (CDF). For both IMPACTS and PLOWS, > 90% of vertical velocity measurements on ~100-meter horizontal scale were < 0.5 m s$^{-1}$ [vertical dashed line in (b)].

Updated version of Figure 13

[Figure]

**Figure 13.** In-cloud vertical velocity measurements taken on level flight legs during IMPACTS and PLOWS on ~100-meter horizontal scale, (a) probability density function (PDF) and (b) cumulative density function (CDF). For both IMPACTS and PLOWS, > 90% of vertical velocity measurements on ~100-meter horizontal scale were < 0.5 m s$^{-1}$ [vertical dashed line in (b)].

**Submitted version of Figure 14**

[Figure]

**Figure 14.** 2D histograms of **(a)** in situ vertical velocity, **(b)** $RH_{ice}$, and **(c)** 2D-S particle counts versus the P-3 distance below CRS cloud

**Updated version of Figure 14**

[Figure]

**Figure 14.** 2D histograms of **(a)** in situ vertical velocity, **(b)** $RH_{ice}$, and **(c)** 2D-S particle counts versus the P-3 distance below CRS cloud

**Submitted version of Figure 15**

[Figure]

**Updated version of Figure (now Figure 16)**

[Figure]

"Samples at air temperatures between −22°C and −8°C represent more than half the samples from both IMPACTS (79,443 100-m samples) and PLOWS (45,927 100-m samples)."

"For IMPACTS and PLOWS, there is a higher incidence of [vertical velocity] values > 0.5 m s$^{-1}$ at air temperature ≤ −22°C (23% and 11%, respectively) as compared to between −8 and −22°C (14% and 5%, respectively)."

"We identified 2305 updraft envelopes ≥ 0.5 m s$^{-1}$ during IMPACTS and 1079 updraft envelopes during PLOWS."

"Median updraft envelope lengths and intensities were 0.28 km and 0.73 m s$^{-1}$ for IMPACTS and 0.24 km and 0.72 m s$^{-1}$ for PLOWS. The mean length of updraft envelopes observed was 0.82 km for IMPACTS and 0.53 km for PLOWS. Overall, 90% of updraft envelopes were shorter than 1.3 km."

"During IMPACTS, 51% of the upward mass flux (Eq. 1) within envelopes meeting our criteria (i.e., not including updrafts weaker than 0.5 m s$^{-1}$) was contributed by updraft envelopes narrower than 3 km. For PLOWS, this value was 72%."

"The distribution of in-cloud vertical velocity measurements during IMPACTS was centered near 0 m s$^{-1}$ (mean: 0.08 m s$^{-1}$, median: 0.04 m s$^{-1}$)"

"less than 15% of the measurements (12.8% for IMPACTS and 7.5% for PLOWS) are capable of lofting snow (≥ 0.5 m s$^{-1}$) on ~100-meter horizontal scales (Fig. 13)."

"We found a broader distribution of in-cloud vertical velocities sampled during IMPACTS than during PLOWS (standard deviations of 0.51 and 0.38 m s$^{-1}$, respectively; Fig. 13a)."

"Just 6.3% of in-cloud 100 m-scale vertical velocity measurements during IMPACTS, and 2.6% during PLOWS, were ≤ −0.5 m s$^{-1}$."

"In-cloud points sampled within 3 km of CRS cloud echo top were nearly twice as likely (13.3% versus 7.3%) to have vertical velocity ≥ 0.5 m s$^{-1}$ than points sampled farther below cloud echo top (Fig. 14a). For downdrafts, 8.7% of points < 3 km below cloud echo top had vertical velocity ≤ −0.5 m s$^{-1}$, compared to 2.6% of points > 3 km below cloud echo top."

"The median $RH_{ice}$ value increases from ~95% at 2 km below cloud echo top height to ~100% in the closest 0.5 km below cloud echo top height (Fig. 14b)."

"Points with $RH_{ice}$ > 105% were observed 13.5% of the time when less than 2 km below CRS echo top, compared to 6.1% of the time when more than 2 km below CRS echo top."'

"– Most updrafts were narrow. 56% in-cloud updraft envelopes exceeding a threshold of 0.5 m s$^{-1}$ were less than 300 m in breadth (median envelope lengths of 0.28 km for IMPACTS and 0.24 km for PLOWS). 90% of updraft envelopes were less than 1.3 km in breadth.
– In-cloud points within 3 km of cloud top were nearly twice as likely (13% versus 7%) to have vertical velocity capable of lofting precipitation-sized ice compared to points sampled further below cloud top.
– The more numerous narrower updrafts contributed more upward mass flux than the scarcer wider updrafts. For updrafts ≥ 0.5 m s$^{-1}$, the majority of the upward mass flux was moved by updrafts < 3 km across.
– Conditions for ice growth (measured $RH_{ice}$ > 105%) were more likely to occur near cloud echo top, while much of the in-cloud cloud volume more than 2 km below echo top had conditions for ice shrinkage (measured $RH_{ice}$ < 95%). Higher number concentrations of particles 0.1 to 2 mm in diameter are more likely to be observed closer to cloud echo top than at lower altitudes."

**Reviewer 1:**

Characterizing the horizontal scales and vertical motions in wintertime precipitating clouds, snow generation in snowbands and generating cells are an important research topic. This study aims to address these areas using a comprehensive data set from three IMPACTS field campaigns (east coast and Midwest US) and the more limited data from the PLOWS (Midwest US) campaign. The main takeaways for me are that the horizontal scales of updrafts are small compared to the broader regions of snowfall, vertical motions are more likely near cloud top, the updrafts are primarily associated with generating cells and snowbands, and the RHi is close to 100%. What are not considered are the weaker updrafts, <0.25 m/s, that might well support much of the ice particle growth. A model could be used to estimate how much of the total water vapor mass upward flux is due to sub 0.5 m/s updrafts.

General Comments

Please comment on whether ice particle growth can largely be ignored outside of the strong updraft regions. Whether there is greater contributions to the total storm precipitation from the strong updrafts versus the weaker but more expansive regions.

Whether vapor depositional growth occurs is dependent on $RH_{ice}$, and the rate of ice particle mass growth with time is linearly related to $RH_{ice}$ (Eq. 9.4 from Rogers and Yau, 1989, p. 160). A motionless particle can grow by vapor deposition if $RH_{ice}$ > 100%. For undiluted parcels, higher w is associated with higher $RH_{ice}$ values, but there is a fair amount of variance in that relationship related to dry air entrainment. Most parcels are diluted even in cumulus clouds (e.g., Blyth, 1993; Blyth et al., 2005; Lasher-Trapp et al., 2005). In the IMPACTS data, the relationship between w and $RH_{ice}$ appears to be quite weak, especially closer to cloud top (new Fig. 15).

In the generating cell level, stronger upward motions and higher RHice were more often observed compared to regions of cloud below the generating cell level, but the joint relationship between stronger upward motions and higher $RH_{ice}$ in a given volume of air is weak. It seems likely that the overturning associated with generating cells near cloud top leads to dry air entrainment and mixing, and so vertical air motions and $RH_{ice}$ are *not* strongly related, especially closer to cloud echo top (new Fig. 15).

[Figure]

**Figure 15.** Joint frequency distributions of 100-m horizontal scale $RH_{ice}$ and vertical velocity (a) < 3 km below CRS echo top, and (b) ≥ 3 km below CRS echo top for the well-coordinated flight-legs during IMPACTS. The color scale in each panel is normalized by the total number of measurements in the panel. The sample size (number of measurements) is indicated in the upper-left corner of each panel.

To help clarify, we removed "Vertical air motions (w)" from the sentence on line 20, and added sentences immediately afterward, so that it now reads:

"Relative humidity (RH) controls where and when hydrometeors are nucleated, and grow or shrink in size. A motionless ice particle can grow by vapor deposition if RH with respect to ice ($RH_{ice}$) > 100%. Saturation vapor pressure is a function of temperature only. Stronger upward air motions, which yield faster decreases in air temperature than weaker vertical motions, are conventionally associated with higher RH values. This works for undiluted parcels, but most parcels are diluted, even in cumulus clouds (e.g., Blyth 1993, Blyth et al. 2005, Lasher-Trapp et al. 2005). Water vapor content is a time-integrated property of an air parcel. A short episode of upward vertical motion, and the corresponding small decrease in air temperature, may not be sufficient to fully counteract previous dry air entrainment and to bring a parcel to saturation."

Further, we have rearranged and reworded some sentences on line 466, which now read:
"In contrast, IMPACTS data sets on $RH_{ice}$ suggest that ice growth is likely to be more episodic than continuous. Ice particles likely grow within small regions of supersaturation with respect to ice for a brief period of time (on the order of minutes) relative to their overall residence time in cloud (on the order of 2 h to fall 8 km).

In order to loft most snow particles, an updraft of at least 0.5 m s$^{-1}$ is needed. Most updrafts stronger than 0.5 m s$^{-1}$ were < 300 m across, with regions of near zero or downward vertical motion in between."

We have added Fig. 15 to the revised manuscript. Correspondingly, we added the following paragraph to Sect. 3.4.2 (line 386):

"One might expect increasing magnitudes of upward vertical air motions to closely coincide with increasing $RH_{ice}$, but the observed data do not show this. The IMPACTS in situ data reveal a negligible relationship between increasing 100 m scale vertical velocity and increasing $RH_{ice}$ (Fig. 15). The lack of a relationship is an important clue that most air parcels within these winter storm clouds are diluted by dry air entrainment to some degree. Comparing the joint distributions between levels within 3 km of CRS echo top and more than 3 km of CRS echo top, the lack of a distinct distribution peak near $RH_{ice}$ ~100% for the parcels that are closer to cloud top suggests that they are more likely to be diluted by dry air than those at lower levels in the cloud. While stronger in-cloud updrafts and higher supersaturations with respect to ice were each more common closer to cloud top (Fig. 15), the presence of one of these conditions at a given location does not imply the presence of the other condition."

We also added the following sentence to the paragraph afterward (line 405):

"While on the one hand, generating cells often contain higher upward vertical air motions, dry air is also more likely to be entrained by the downward branches of the overturning near cloud top, blurring the relationship between 100-m scale vertical air motion and RH."

In the Conclusions, we added this bullet point on line 454:

The negligible relationship at 100-m horizontal scale between increasing vertical air motions, and increasing RHice ≥ 105% implies that most air parcels within winter storms are diluted by dry air entrainment.

And we rephrased and added to the following sentences on line 484:

"Yielding and maintaining conditions of supersaturation with respect to water requires stronger updrafts than does maintaining supersaturation with respect to ice. Dry air entrainment leads to dilution of air parcels such that there is a negligible relationship between 100-m scale vertical air motion and RH for the same place and time."

Given the vertical velocity measurement accuracy, you are not measuring <20 cm/s updrafts. These can be significant in providing an upward flux of moisture for particle growth downwards.

We cannot characterize weak updrafts and downdrafts with the observations because of the limit in instrument accuracy. Updrafts with weak velocities < 20 cm/s are outside of the scope of what we can analyze with the observations. But, we take the point that it is important to keep the measurement uncertainty in mind when interpreting our results. On line 348, we therefore added:
"(i.e., not including updrafts weaker than 0.5 m s$^{-1}$)".
In the bullet point on line 448, we added:
"For updrafts ≥ 0.5 m s$^{-1}$".

Lines 220-221. precipitation size ice particles. According to the AMS Glossary of Meteorology, frozen precipitation: Any form of precipitation that reaches the ground in frozen form, that is, snow, snow pellets, snow grains, ice crystals, ice pellets, and hail. With 0.5 m/s threshold, how much precipitation would you be missing (percentage wise).

We chose 0.5 m s$^{-1}$ as a "lower bound" on the median fall speed of precipitation size ice, i.e., most precipitation size ice particles have terminal fall speeds > 0.5 m s$^{-1}$.

The classic Locatelli and Hobbs (1974) paper measured few if any particles with fall speeds less than 0.5 m s$^{-1}$ (Fig. R1).

[Figure]

Fig. 4. Ranges of maximum dimensions, masses, and fall speeds of the solid precipitation particles observed in the present study.

Figure R1. Figure 4 from Locatelli and Hobbs (1974) with red line added at 0.5 m s$^{-1}$.

Figure 6 in Fitch et al. (2021), given below as Fig. R2, presented distributions of precipitation size ice particle fall speeds, measured at the surface using a Multi-Angle Snowflake Camera (MASC), for light wind speed conditions in the Arctic. Most unrimed aggregates had fall speeds > 0.5 m s$^{-1}$, and the fall speed distributions for the moderately rimed and heavily rimed (graupel) particles, which are denser than aggregates, were shifted toward faster fall speeds.

Especially in high turbulence and at low temperatures (< -13°C), there is almost no relationship between ice precipitation particle size and fall speed (Garrett and Yuter, 2014), as we mention on lines 69-71.

Lachapelle et al. (2024, JAMC) measured precipitation particle fall speeds at the surface using a laser-optical disdrometer. From their Figure 14, given below as Fig. R3, the vast majority of snow (SN), ice pellets (PL), and refrozen wet snow (RWS) particles had fall speeds exceeding 0.5 m s$^{-1}$.

The distribution of ice precipitation particle fall speeds as a function of size and mass is not well-constrained. Factors such as horizontal wind speeds, turbulence, particle shape, and degree of riming can have substantial impacts on fall speeds. It is not possible to measure particle fall speed from a fast moving airplane. We cannot say exactly what proportion of precipitation size ice (> 0.2 mm diameter) particles fall slower than 0.5 m s⁻¹, but based on surface-based measurements of fall speed in the literature and inferring more turbulent and colder environments aloft it is almost certainly less than 50%.

[Figure]

*Figure R2 (Figure 6 from Fitch et al. 2021). PDF estimates for shielded MASC fall speed $v_p$ for different horizontal wind speed categories <= 1.5 m/s, <= 1.0 m/s and <= 0.5 m/s and hydrometeors divided into three riming classes: sparsely rimed aggregates, moderately rimed, and rimed.*

[Figure]

*Figure R3 (Figure 14 from Lachapelle et al. 2024 with red line added at 0.5 m s$^{-1}$). Time series of fall speed during the documented precipitation type transitions on (a) 19 Nov 2019 ($D_{th}$ = 0.87), (b) 12 Jan 2020 ($D_{th}$ = 1.75), (c) 27 Feb 2020 ($D_{th}$ = 0.87), and (d) 23 Nov 2019 ($D_{th}$ = 1.25). Ice pellet occurrences all show slow particles falling between 1 and 2 m s$^{-1}$. The time series show the counts of particles > $D_{th}$. This allows visualization of the slower particles' fall speed distribution that would otherwise be hidden behind the high number of smaller and slower particles. Fall speeds between 1 and 2 m s$^{-1}$ have been highlighted in gray. Each event is divided into intervals associated with one or two precipitation types that were reported by manual observers. The time intervals during which manual observers identified ice pellets as the main type of precipitation are indicated with a pink label.*

Lines 423-424 These results show that the types of winter storms sampled by IMPACTS and PLOWS (extratropical cyclones, e.g., Nor'easters, Alberta clippers, and Great Plains cyclones) contain mostly weak vertical motions incapable of lofting precipitation-size ice.).
Please comment on the size range of particles that may be lofted at 0.5 m/s. This would provide an upper limit for the sizes that may be lofted.

See response to previous comment regarding Lines 220-221.

We added the following sentences in the introduction on line 66:
"The fall speeds of individual ice crystals further depend on the variations among ice crystal shapes within a broad category like aggregates (e.g., Vázquez-Martín et al., 2021). The classic Locatelli and Hobbs (1974) paper measured very few fall ice particle speeds < 0.5 m s$^{-1}$ (their

Figure 4). More recent work by Lachapelle et al. (2024) also observed few fall speeds in snow < 0.5 m s$^{-1}$ (their Figure 14)."

I suggest adding a figure showing the distribution of temperatures sampled for IMPACTS and PLOWS
Figure 10 shows these distributions.

Should the measured updrafts be called flight segments (which they are) or envelopes (how do you know)?
We define the continuous in-cloud vertical velocity measurements > 0.5 m s$^{-1}$ as "updraft envelopes" rather than as "updrafts" because the aircraft measurements do not capture the entire updraft region, as discussed in Sect. 2.5. "Flight segments" could be confused with "flight legs".

Are the TAMMS inlets subject to icing and therefore error?
If the TAMMS pressure ports are iced, they are totally blocked, and the data is flagged as missing.

Lines 450-456. This is superfluous and should be removed.
We respectfully disagree as ventilation is something that is relevant, not well understood for complex real ice shapes, and not accounted for in current numerical models, so it is worth mentioning.

Minor Comments

37 gravity waves
We have reworded this sentence and added "gravity waves." It now reads:
"Processes such as, e.g., buoyancy, turbulence, gravity waves, and vertical pressure gradients can yield vertical motions of several meters per second or more at scales less than a few kilometers."

44. timescale of snow. From when. From its nucleation through to ...?
We are not counting the time that the particle is cloud-size ice and not falling. We mean the amount of time starting from the moment a particle reaches precipitation-size (begins to fall relative to the air) and the moment it reaches the surface. We have restructured this paragraph to try to make this clearer, also in response to a comment by Reviewer 2:
"Microphysical properties of hydrometeors are time-integrated. The "microphysical pathway" is the succession of mass changes a hydrometeor undergoes as a function of the sequence of relative humidity (RH) and temperature environments that the hydrometeor encounters as it moves along its trajectory through the storm. The length of time between when a cloud particle reaches precipitation size and begins to fall out, and when the particle reaches the surface has been called the cloud "delay time" and the "residence time" by different authors (e.g., Feingold et al., 1996; Barstad and Smith, 2005; Smith, 2006; Janiszeski et al., 2023). We will use residence time in this article to refer to the time for precipitation-size ice (diameter > 0.2 mm) to

fall out which excludes the time the particle spends as cloud-size ice and is not falling out. The spatial distribution of surface precipitation is highly sensitive to the residence time (e.g. Smith, 1979; Colle and Mass, 2000; Colle and Zeng, 2004; Colle et al., 2005; Lackmann and Thompson, 2019). When residence time is increased, the lengthening of the snow particle trajectory yields more time for advection by horizontal winds and for particle growth and/or shrinkage processes to occur prior to the particle reaching the surface."

47. begins to fall. It's always falling during its growth, because it has a terminal velocity. It may not fall through the updraft. So, fall>fallout

51. "fall" to "fallout" Fall speed should be "terminal velocity". fall speed is the net (updraft velocity-terminal velocity).
We have changed "fall" to "fall out" in these two sentences. We also changed "fall speed" to "terminal velocity" where appropriate.

197. Should be University of North Dakota and NCAR
We have corrected this. UND and NCAR operated the probes, some of which came from SPEC. This sentence on line 210 now reads:
"During IMPACTS, groups from the University of North Dakota (UND; Delene and Poellot, 2022) and National Center for Atmospheric Research (NCAR; Bansemer et al., 2022) operated cloud sampling instruments and quality controlled the data."

202. Fig. 5. Note in the text that this figure will be discussed in more detail later
We have corrected the ordering of figures to reflect the order in which they appear in the text.

Fig. 5 If you're using all channels of the 2D-S probe, because of uncertainty in the sample volume of the particles in the bins for the small particles, it's an overestimate because some of the particles may be drops, shattering affecting the concentration and uncertainties in the sample volume for the small particles.
We used the "total number concentration" variable in the 2D-S data, which only includes particles larger than 100 μm, thus excluding the smallest size bins.

314. RHWater >100% is extremely unlikely if not impossible, especially at these vertical velocities.
The likelihood of $RH_{water}$ > 100% is not particularly relevant here, as this paragraph is simply describing the conditions required for different ice particle growth modes. Regardless, observations of branched ice particles and liquid water droplets in these clouds imply that $RH_{water}$ > 100% must have occurred somewhere at some point in time, even if only in the immediate vicinity of particles (e.g., ventilation effects).

379-380. Saturation with respect to water not ice
The discussion here is about ice particles, not liquid water droplets, so we respectfully disagree.

399-407 This is an excellent paragraph

444. and aggregation

We are not aware of much if any published observational work on whether cloud size ice particles grow to precipitation size by aggregation, i.e., whether aggregates contain cloud-size ice particles. Until recently, optical probes were not good enough to address it. For example from these 2DC images one cannot tell much about the constituent particles.

[Figure]

Based on our analysis of PHIPS and MASC observations (which are good enough to discern individual component crystals), most aggregates seem to be jumbles of precipitation-size (D >= 0.2 mm) particles rather than of smaller cloud ice but we have not looked at this systematically.

(to left) MASC images courtesy of Tim Garrett.

(to right and below) Example PHIPS optical probe image of an aggregate that clearly shows the component ice particles, magenta scale bar is 0.5 mm long.

[Figure]

Aggregate of rimed columns

Figure 11 multiple ice growth. A better description is needed.

From Hueholt et al. (2022): "at low ice supersaturations and most temperatures, vapor deposition yields multiple ice shapes (polycrystals, plates, irregulars, compact crystals, short columns, and equiaxed crystals)." We have reworded this part of the figure caption to reflect this:

"Depending on the RH, polycrystalline growth can occur at temperatures < -22°C. Tabular, branched, or side branched growth can occur between -22 and -8°C. Columnar growth can occur between -8 and -4°C. At most temperatures, multiple ice shapes form at low supersaturations with respect to ice, including polycrystals, plates, irregulars, compact crystals, short columns, and equiaxed crystals  (Bailey and Hallett, 2004, 2009; Hueholt et al., 2022)."

Figure 11: <-22C

Thank you, this has been fixed.

450-456. This is  superfluous and should be removed.

We respectfully disagree, as discussed above.

463-470. This part of the paragraph is superfluous and should be removed.

We respectfully disagree. Ice particle images may yield important insights into the microphysical pathways encountered by hydrometeors which cannot be gained through analysis of the vertical velocity and humidity data alone.

**Reviewer 2:**

The authors address an important issue of ice particle generation and growth in cloud top generating cells found in winter storms sampled using observations from the IMPACTS and PLOWS field campaigns. The authors show that narrow updrafts of >0.5 m s$^{-1}$ associated with such generating cells are primarily responsible for ice particle generation, growth, and particle lofting within the cloud top layer while ice particle size is likely constant or decreasing beneath in most cases. The authors also do a great job explaining their findings in the context of previous work on these winter storms and how previous understandings were either incorrect or limited. The overall manuscript is very well written with clear findings and well produced figures to support and detail the main findings which make a significant contribution to the scientific knowledge for ice particle formation and growth in such storms. I have only some minor comments for the authors to address:

Line 37: I would add that gravity waves can also yield vertical motions.

We have reworded this sentence and added "gravity waves." It now reads:
"Processes such as, e.g., buoyancy, turbulence, gravity waves, and vertical pressure gradients can yield vertical motions of several meters per second or more at scales less than a few kilometers."

Lines 42 – 54: Maybe start with explaining residence time first in this paragraph (after the first sentence) then explain the changes or "microphysical pathway" the hydrometeor undergoes. I think you can fold the paragraph from lines 51-54 into the paragraph above. That way you can remove the "timescale of snow falling to the surface" and just use residence time. Just a suggestion not a must fix.

Thank you for this suggestion. We have restructured this paragraph to try to make this clearer, also in response to a comment by Reviewer 1:
"Microphysical properties of hydrometeors are time-integrated. The "microphysical pathway" is the succession of mass changes a hydrometeor undergoes as a function of the sequence of relative humidity (RH) and temperature environments that the hydrometeor encounters as it moves along its trajectory through the storm. The length of time between when a cloud particle reaches precipitation size and begins to fall out, and when the particle reaches the surface has been called the cloud "delay time" and the "residence time" by different authors (e.g., Feingold et al., 1996; Barstad and Smith, 2005; Smith, 2006; Janiszeski et al., 2023). We will use *residence time* in this article to refer to the time for precipitation-size ice (diameter > 0.2 mm) to fall out which excludes the time the particle spends as cloud-size ice and is not falling out. The spatial distribution of surface precipitation is highly sensitive to the residence time (e.g. Smith, 1979; Colle and Mass, 2000; Colle and Zeng, 2004; Colle et al., 2005; Lackmann and Thompson, 2019). When residence time is increased, the lengthening of the snow particle

trajectory yields more time for advection by horizontal winds and for particle growth and/or shrinkage processes to occur prior to the particle reaching the surface."

Line 173: Reword end of sentence. "within these types winter storms available" doesn't make sense.

Thank you for catching this. We have reworded the sentence to say:
"The vertical air motion distributions sampled during these projects are the best evidence of natural conditions within these types of winter storms."

Line 174: Just say "warm and/or occluded fronts" and remove "air mass boundaries".

We have made this change. The sentence now reads:
"Warm and/or occluded fronts are often found in the northwest and northeast quadrants, which are generally associated with frontogenesis and strong vertical wind shear (which may be sufficient for Kelvin-Helmholtz instability)."

Line 202: The reference to Fig. 5 makes it appear as if the figures are out of order. It would be good to add that this figure will be discussed later on in the text.

We have reordered the figures so that they now appear in the correct order of their first mention in the text.

Lines 374 – 379 summarizes the section very well

The discussion section is very well done

Lines 450 – 456: Not a must fix but I'm not sure this information is really needed for this paper.

We view ventilation as something that is relevant, not well understood for complex real ice shapes, and not accounted for in current numerical models, so it is worth mentioning.

Lines 463 – 472: Good description of future work possibilities for the IMPACTS data.

Figure 11 caption: should be -22 C not 22

Thank you, this has been corrected.

**Reviewer 3:**

Overarching comment:

This paper presents analysis of aircraft vertical motions across two field campaigns, putting these observations in the context of the thermodynamic, kinematic, and structural features of the cyclone. The paper is well-written, has clear figures, and clearly lays out its findings.

The definition of an updraft is fairly fundamental to the analysis of the paper, and something I find a bit confusing. The choice of 0.5 m s$^{-1}$ is certainly defensible, but could use more explanation. What exactly is the uncertainty of the instrument? I'm not sure if the paper mentions it, but even if it does, putting it near Ln. 219 would help the reader understand why the value was chosen. Additionally, discussing these values in context of microphysics, particularly terminal velocity, could use a bit of clarity. Often, terminal velocity is discussed alongside vertical velocity due to the use of radar velocity measurements. Based on the rest of the paper, I think the point of the terminal velocity comparison is to note regions where the vertical air motion is sufficient to loft particles (that is, the particles themselves are moving upwards). I think making this clear in that section will make it more clear why the line is drawn where it is.

Thank you for pointing this out; we neglected to mention the instrument uncertainty (±0.5 m s$^{-1}$) in Sect. 2.1. We have added sentences to that section to clarify, first on line 138:
"The uncertainty in TAMMS vertical velocity measurements is ±0.5 m s$^{-1}$."

And on line 162:
"As with the IMPACTS data, the PLOWS vertical velocity measurements have roughly ±0.5 m s$^{-1}$ uncertainty."

We also reworded this sentence in Sect 2.4, on line 233:
"We use a similar method to Yang et al. (2016) to identify updraft envelopes, but we have a measurement uncertainty in vertical velocity of ±0.5 m s$^{-1}$, so we use a 0.5 m s$^{-1}$ threshold for updrafts."

After this sentence, we already had a sentence stating:
"This threshold also represents a lower-end estimate for the terminal velocity of precipitation size ice particles (Garrett and Yuter, 2014; Fitch et al., 2021)."

A related comment is that the paper discusses velocities approaching zero several times. Especially with the context of the discussion of vertical velocity scales earlier in the paper,

it's important to note the difference between approaching zero, but having a statistically significant difference from zero (that is, one can say that the mean is either up or downward motion), and approaching zero, where the mean could be zero. For instance, in Fig 1c, the caption notes that the maxima and minima approach zero, but values are still generally above zero (upwards motion). I suggest considering this distinction when the paper discusses velocities approaching zero. When averaged over larger scales, is the motion still upwards?

Given the measurement uncertainty (0.5 m s$^{-1}$), we cannot definitively distinguish values between 0.0 and 0.5 m s$^{-1}$ from 0 m s$^{-1}$.

For this particular leg, mathematically the 100-sec average velocity is slightly positive.

The Fig. 1 caption and associated text were the only points where we discuss vertical velocity approaching zero. We were only referring to the decreased magnitude of local extrema in vertical velocity. We reworded this part of the Fig. 1 caption to clarify:
"As the averaging period increases, the magnitudes of maxima and minima in vertical velocity become smaller, and the portion of the flight leg where vertical velocity ≥ 0.5 m s$^{-1}$ decreases, and there is a decreasing portion of the flight leg where vertical velocity is outside of the measurement uncertainty range of ±0.5 m s$^{-1}$."

We also reworded this sentence in Sect. 1 (line 44):
"Figure 1 uses vertical velocity measurements from a single flight leg during IMPACTS to illustrate that as the horizontal scale of measurements is increased from 0.1 km to 10 km, the maxima and minima in vertical velocity become smaller in magnitude."

General comments:

Unless I missed something, it appears the manuscript references figures out of order. This makes it confusing for the reader to follow.

We have reordered the figures so that they now appear in the correct order of their first mention in the text.

Ln 16: The trajectories are not the only control, but are an often-neglected control.

We have added the word "partly" here, so that the sentence now reads:
"The 3D trajectories of precipitation-size ice particles through winter storms partly control the amounts and spatial distribution of surface snowfall accumulation."

Ln 196: What does "handle" mean here - did they QC the data?

Groups from UND and NCAR operated the instruments and QCed the data. We reworded this sentence to clarify:
"During IMPACTS, groups from the University of North Dakota (UND; Delene and Poellot, 2022) and National Center for Atmospheric Research (NCAR; Bansemer et al., 2022) operated cloud sampling instruments and quality controlled the data."

Ln 208-210: If QC was performed, was the PLOWS data handled the same way?

Archived data for PLOWS were QCed prior to being posted. See archive for reference.

Ln 239-240: I think the spirit of this comment is correct, but another flight hazard - icing - will discourage flying in the largest updrafts (in certain temperature ranges) due to strong updrafts potentially reaching liquid saturation.

We added the following sentence on line 256 to address icing:
"Regions with severe icing, which might have strong upward motion, were also avoided during research flight missions."

Ln. 273: I think there are good reasons to prefer a reanalysis vs a higher resolution NWP model here, but I would suggest the manuscript spells out for the reader why a reanalysis was chosen.

We reworded and added a sentence to line 288 to clarify that the ERA5 grid spacing is adequate for resolving the synoptic-scale environment:
"ERA5 data are output on a 0.25° grid (~25 km) globally at 1 h intervals. While this is coarser than most operational weather models, the ERA5 grid spacing is adequate for resolving the synoptic-scale environment. Reanalysis assimilates quality-controlled observations that are delayed and not available for use operationally."

Ln. 295-297: Here's an example of where choice of context may matter. Does ERA represent the environment sufficiently enough that the omega in Fig. 6 is on a scale that's useful to the story - think back to the discussion of the scales of updrafts earlier.

We use ERA5 output to characterize the large (25 km) scale coarse resolution context of the IMPACTS observations. We have added text reiterating the scale of the ERA5 output in Sect 3.1, on line 312:

"Based on ERA5 reanalysis (Fig. 7), the flight leg was to the northwest of and near a band of frontogenesis > 1 K (100 km)$^{-1}$ h$^{-1}$ at 700 hPa. There was negative 25 km-scale frontogenesis (i.e., frontolysis) present at the P-3 flight level with modest 25 km-scale upward motion."

On line 318:
"This flight leg was to the northwest of and near a band of weak 25 km-scale frontogenesis at 700 hPa. Along flight level, weak 25 km-scale frontogenesis was present (Fig. 8a) and 25 km-scale upward motion is indicated just above flight level."

In Sect. 3.2, on line 321:
"The distributions of ERA5 frontogenesis and omega for all the IMPACTS flight legs utilized in this study (Fig. 9) show that the most commonly sampled environments had 25 km-scale frontogenesis near 0 K (100 km)$^{-1}$ h$^{-1}$ and 25 km-scale omega near -0.4 Pa s$^{-1}$. Strong frontogenesis and strong upward motions on the 25 km scale are outliers."

Ln. 306-308: Could there be a sampling strategy difference, type of storm sampled difference (e.g. more cold Clipper type systems during PLOWS), or an altitude difference that explains this?

There does appear to be a difference in the altitudes sampled by IMPACTS and PLOWS. PLOWS tended to sample more often at higher altitudes, in particular above 4500 m, compared to IMPACTS (Fig. R5), which could explain some of the differences in the temperature sampling. The colder climatological surface temperatures in the Midwest compared to the northeastern US are still likely relevant to the difference in the temperature distributions sampled by IMPACTS and PLOWS, and the common storm types in each region are a part of that climatological temperature difference.

[Figure]

*Figure R5. Distributions of in-cloud altitudes above sea level (ASL) sampled during (a) IMPACTS and (b) PLOWS.*

We reworded the text on line 323 to take the different altitude sampling into account: "The air temperature distributions of the in situ samples are shifted to higher temperatures for IMPACTS as compared to PLOWS (Fig. 10). PLOWS tended to sample higher altitudes (> 4500 m above sea level) more often compared to IMPACTS (not shown), but climatologically winter surface temperatures in the Midwest US also tend to be lower than in the northeastern US."

Ln. 344-346: What is the paper defining the "entire storm" to be? I think the general strategy for sampling these storms is to go from one edge of the precipitation echoes to the other (insofar as one can do so with the constraints of aviation navigation). These legs are likely not on the extremities of the precipitation region, so this statement is probably true to a certain extent, but how unrepresentative are these values for the comma-head? Stratiform precipitation in cyclones? Etc.

IMPACTS flight legs targeted "areas of interest" with likely snow growth, e.g., bands of locally enhanced radar reflectivity. The entire storm includes regions where snow growth is less likely. That includes the edges of the precipitation region as mentioned, but also broader stratiform precipitation regions with low radar reflectivity which were a lower priority during IMPACTS flights. We cannot quantify how much we are likely overestimating the proportion of the storm volume with vertical velocity ≥ 0.5 m s$^{-1}$.

We have added a bit of text to the sentence on line 363 to specify that flight missions targeted regions of locally enhanced radar reflectivity:

"Because the research flights during IMPACTS and PLOWS targeted regions of likely snow growth (e.g., regions of locally enhanced radar reflectivity), we expect the proportion of aircraft samples with vertical velocity ≥ 0.5 m s$^{-1}$ to be an *overestimate* compared to the proportion of the entire storm volume with vertical velocity ≥ 0.5 m s$^{-1}$."

Discussion/conclusions section:

I generally agree with the points here. But I want to add a couple of thoughts that I had while reading that these sections don't really address. Some of this may be "future work" rather than items that can be addressed with these results, but they are questions raised by the emphatically-worded conclusion in Ln 406-407.

1. Let's say that the point here, that cloud top generating cells are where all the action is at, is correct. Why do operational NWP models - which neither resolve nor parameterize generating cells - manage to get usable forecasts for these storms?

   It depends on what one defines as usable. Often, quantitative precipitation forecasts (QPF) for snow have very large errors (> 100%), large enough that improving this is a key priority for NOAA (National Weather Service, 2018). We added this sentence to Sect. 4, on line 434:
   "Those errors may then cascade into quantitative precipitation forecasts which often have uncertainties exceeding 100% for snow (Novak et al., 2008, 2014, 2023; Greybush et al., 2017; National Weather Service, 2018)."

2. If upward vertical motions and cloud saturation happen near cloud top - 8 km up or more at times - does this imply conditions at the ground are essentially decided an hour plus in advance due to fall time (assuming negligible loss due to sublimation/virga processes)? This would have significant decision support implications!

   Interesting insight. Yes, it would mean that the "snow on the way" mostly is based on what is going on near cloud top and the wind shear structures the particles encounter on the way down. A practical issue is that operational weather radars usually have insufficient spatial resolution to resolve ~1 km spatial scale generating cells. Shear profile info could be estimated with VADs or wind profiler data. We added these sentences to the Conclusions, on line 480:
   "This implies that conditions on the ground during snow storms could be largely determined an hour or more in advance, given sufficient knowledge of cloud top structures and the shear profile. However, operational weather radars usually have

insufficient spatial resolution to resolve ~1 km spatial scale generating cells, and information on the shear profile is largely limited to radar-derived velocity azimuth displays (VADs) or data from wind profilers."

3. Given that the processes around generating cells and their updrafts are of a small enough scale, what component of the findings would be easiest to apply to representing generating cells in models - adjusting microphysics parameterizations to grow particles closer to cloud top, or adjusting thermodynamic profiles to match the observations (highest RH near cloud top)? The paper touches on this at the end of the conclusions, but it might be interesting to compare observations vs the reanalysis data.

Addressing *how* best to represent generating cells in forecast models with a range of grid spacing values is beyond the scope of this paper. We added this sentence on line 438:
"*How* best to do this is a worthy topic for future research."

---

## Referee Report (RR1)

Review of "In-cloud characteristics observed in US Northeast and Midwest non-orographic winter storms with implications for ice particle mass growth and residence time", by Allen and coauthors, egusphere-2024-3808

The authors have addressed the reviewer comments well and have made the relevant changes to the manuscript. After having read the revised manuscript, I believe it is in excellent shape and will be a significant contribution to the scientific knowledge of ice particle formation and growth in such storms. I look forward to seeing this manuscript published.